

**Understanding the soil loss at the permanent gully headcut area in the Mollisols region**
**of Northeast China**
Chao Ma[1], Shoupeng Wang[1], Dongshuo Zheng[1], Yan Zhang[1], Jie Tang[2], Yanru Wen[3], Jie Dong[4]
1. School of Soil and Water Conservation, Beijing Forestry University, Beijing 100083, PR China.
2. Advanced Institute of Natural Sciences, Beijing Normal University at Zhuhai, Zhuhai 519087, China
3. Institute of Agricultural Resources and Regional Planning, Chinese Academy of Agricultural Sciences, Beijing
100081, China
4. Civil and Environmental Engineering Department, Clarkson University, NY, 13699, USA.
Corresponding Author: Professor Chao Ma, sanguoxumei@163.com
**Abstract:** The development of permanent gullies can trigger both gravitational mass-wasting on over-steepen slopes
and water erosion on the channel bed. This hydrogeomorphic process is typically driven by the hydrology process
of the headcut area and the hydro-mechanical response within the soil mass. In this study, erosion intensities were
observed at the headcut area of two permanent gullies in the Mollisols region of Northeast China during the rainy
and snow-melting seasons. To understand water storage capacity and leakage process, as well as the suction stress
level during the rainy and snow-melting seasons, critical parameters such as soil moisture, temperatures, and
precipitation amounts were investigated. This analysis also incorporated the effects of pore water pressure rising and
dissipation properties, and hydro-mechanical properties of Mollisols. The Mollisols at the interrupted headcut area
of gully No. II exhibited a higher ratio and proxy of pore water rising and dissipation than those at the uninterrupted
headcut of gully No. I. Moreover, the combination of soil and water characteristic curve along with the hydraulic
conductivity function (HCF) indicate that the Mollisols of gully No. II has relatively higher air-entry pressure and
saturated hydraulic conductivity during wetting and drying cycles than those of gully No. I. The headcut area of
gully No. II exhibited rapid water infiltration and leakage responses during rain events, with high capacity in the
water storage during torrential rain, rainstorm, and snow-melting season. Overall, the absolute suction stress within
the Mollisols of gully No. II was lower than that of gully No. I, which could lead to high erosion intensity on the
over-steepen slope. Importantly, we provided new evidence that the area erosion intensity of gravitational mass-
wasting on the over-steepen slope was closely related to the soil suction stress level. Additionally, we observed a
correlation between the erosion intensity of the gully bead near the headcut and the soil water storage. The findings
of this study significantly deepen our understanding of the physical process of permanent gully development in the
Mollisols region. We provide important insights that the accuracy of the Universal Soil Loss Equation could be
improved by accounting for the effects of soil water storage pattern and soil suction stress status.
**Key words: Gravitational mass-wasting; Soil water characteristic curve; Areal erosion intensity**
**1 Introduction**
The gravitational mass-wasting process refers to the downward movements of rock, regolith and/or soil, caused
by gravity, along the sloping top layers of the Earth's surface (Evans, 2004; Allaby and Michael, 2013). There are
four types of mass wasting based on the speed of the material's movement downslope and the level of moisture.
These include falls and avalanches, landslides, flow, and creep (Bierman and Montgomery, 2014). They often occur
in various sizes with undetermined failure planes, impacted by hydrology and hydro-mechanical responses (Stein
and LaTray, 2002; Rengers and Tucker, 2015; Allen et al., 2018). In the over-steepen slope of the permanent gully,
gravitational mass-wasting processes involve soil debris free falling due to bed undercutting driven by intensive
channelized flow or persistent high soil moisture conditions (Harmon and Doe, 2001). From the unsaturated soil
mechanics, a high potential of occurrence or strong intensity of erosion for gravitational mass-wasting when the soil
mass is in low suction stress status (Lu and Godt, 2013). Note that both rain and snow-melting water infiltration



trigger low soil suction stress. It is still unclear whether the erosion intensity of gravitational mass-wasting process
corresponds to the state of soil suction stress during these two stages. The first case occurred on the condition that
the over-steepen slope lost support from debris deposits, while the second involved persistent low soil suction stress.
Permanent gullies initiate in locations where concentrated flow can erode and move sediment (Kirkby and
Bracken, 2009), and expand at the over-steepen slopes when gravitational mass-wasting process occur following
instant or constant water infiltration (Poesen et al., 2010; Tebebu et al., 2010). The erosion intensity of gravitational
mass-wasting, one of the most important factors in the development of permanent gullies, could be determined by
the topographical threshold and volumetric retreat rate of gully headcut (Svoray et al., 2012; Guan et al., 2021; Zare
et al., 2022), gully length-area-volume relationship (Li et al., 2015 and 2017), as well as their function with upstream
drainage area and rainy days in different environments (Hayas et al., 2019). In fact, the erosion rate of permanent
gully is largely influenced by the hydrological factors (Gómez-Gutiérrez et al., 2012). These include the flow rate
and total water volume, which are determined by critical topographic conditions, rainfall intensity and volume, and
hydro-mechanical properties of soil mass. Additionally, these soil properties are further impacted by the land use,
plant roots, soil texture and structure. Currently, most studies on permanent gullies have primarily concentrated on
the gully headcut retreat and topographic threshold conditions (Torri and Poesen, 2014; Vanmaercke et al., 2016).
The hydrology and the hydro-mechanical response of the gully heads to water infiltration and their relationship with
the erosion intensity of gravitational mass-wasting remain unknown. In natural conditions, water infiltration can
occur due to either rain events or the snow/ice melting. The infiltration rate greatly depends on both the amount and
intensity of precipitations, leading to water storage. However, the level of stored water notably varies due to the
precipitation pattern and the melting rate of snow/ice. In general, during the snow/ice melting season, the duration
of water infiltration persists longer than the rain events as a result of prolonged soil saturation with an extended
period of low soil suction stress status. This phenomenon may generate more gravitational mass-wasting and higher
erosion intensity. However, rain events typically generate intensive channelized flow, which erodes over-steepen
slopes and triggers gravitational mass-wasting. Therefore, it is challenging to compare erosion intensity in two
different seasons, but this issue could be addressed by considering the associated hydrological and hydro-mechanical
responses of the over-steepen slope near the gully head.
In the Mollisols region of Northeast China (MEC), over 295,000 permanent gullies have developed since 1960
(Li et al., 2013; Dong et al., 2019), and the gravitational mass-wasting processes have caused rapid gully widening
due to over-farming and a lack of maintenance (Wang et al., 2009). Various studies focused on hydrological
processes affecting the ephemeral gully development and volume disparities caused by rain/snow patterns (Tang et
al., 2022; Jiao et al., 2023), tillage practice (Xu et al., 2018; Li et al., 2021), and morphological characteristics (Zhang
et al., 2016). Note that permanent gully poses a greater threat for cropland than the ephemeral gully, because the soil
loss caused by permanent gully erosion can be as high as 50~65% of the total loss in northeast China (Zhang et al.,
2022). Importantly, recent studies have found that a relatively high area increasing ratio is impacted by the
combination of permanent gullies with cropland use, large ridge orientation angle, and sunny orientation (Li et al.,
2016; Liu et al., 2023). Tang et al (2023) proved important evidence of the rainfall threshold for permanent gully
development and found that the maximum value of 3-day accumulative rainfall best explained permanent gully bed
erosion, and the accumulative value of erosive rainfall best accounted for the gravitational mass-wasting. However,
the gravitational mass-wasting on the over-steepen slope of the permanent gully can occur either in the rainy season
or the snow-melting season (Zhang et al., 2020; Zhou et al., 2023). Currently, the hydrology response and the hydro-
mechanical properties of the over-steepen slope in the two seasons have been never documented and the associated
erosion intensity of gravitational mass-wasting was poorly understood.
The hydrological response and hydraulic properties of the over-steepen slope are paramount to know the
gravitational mass-wasting (Sidle et al., 2017). In MEC, though the duration of the snow/ice melting season is shorter



than the rainy season (Wang et al., 2021; Fan et al., 2023; Went et al., 2024), the time lasting for snow-melting water
or rainy water infiltration, the water storage and leakage process are significantly different. Firstly, the water storage
may exceed the water leakage in the snow/ice melting season because of continuous melted water infiltration and a
limited water leakage path. Secondly, rain infiltration during the summer season temporally increases and decreases
afterward once the rain event ceases and water leakage starts. Meanwhile, stored water greatly depends on the rainfall
pattern (Farkas et al., 2005; Xu et al., 2018). Therefore, the duration of low soil-suction stress status, e.g., the high
soil moisture stage, greatly differs in the two seasons. Another effect is the channelized water during intensive
rainstorms (Wen et al., 2021), which may erode the bed and result in gravitational mass-wasting. Therefore, the
erosion intensity of gravitational mass-wasting may coincide with the soil suction stress status in the snow/ice
melting season, while the coincidence may not exist in the rainy season.

In the present study, we utilized a combination of field soil moisture-temperature-rainfall observations, pore

water pressure dissipation and hydraulic conductivity analysis of over-steepen slope materials, soil suction stress
estimation, water storage-leakage, and topographical changes to explain the erosion intensity of gravitational mass-
wasting relating to hydrology and hydro-mechanical response. Particularly, the headcut area of two permanent
gullies were selected in this work. One headcut region experiences no human activities, while the other does. The
Mollisols at these two headcuts have different hydraulic conductivity, thus the erosion intensity of gravitational
mass-wasting, hydrology, and hydro-mechanical response in snow-melting and rainy season could be distinctive.
Additionally, the soil water characteristic curve (SWCC), hydraulic conductivity function (HCF), and the pore water
pressure dissipation of the Mollisols were compared. Then, we highlighted the importance of soil water storage and
drainage, and suction stress in during the rainy and snow-melting seasons. The results of this work will deepen our
understanding of permanent gully expansion from both classical mechanics and the state of stress perspectives.

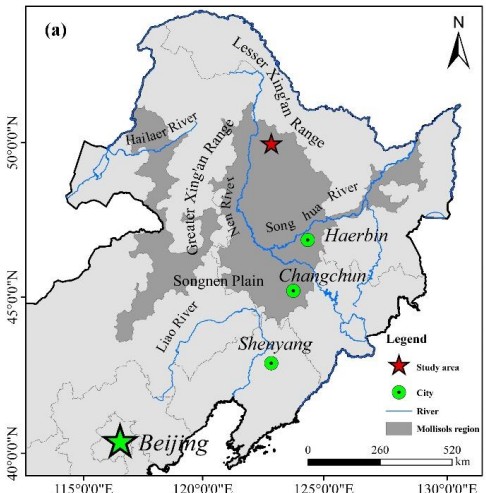

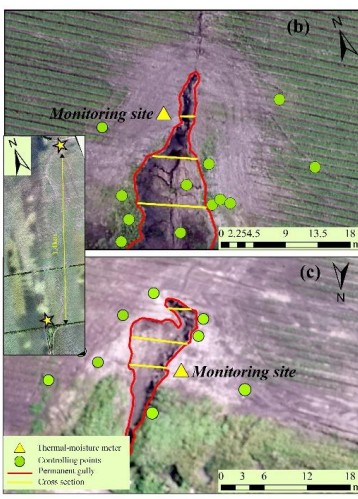

**Fig. 1.** Location of the two permanent gullies in the Mollisols region of northeast China. **(a)** The red star marks
observation site in the study area (from ESRI). **(b)** Monitoring sites and ground controlling points at permanent
gully No. I. **(c)** Monitoring sites and ground controlling points at permanent gully No. II. (background of **a** is
from ESRI; areal maps of **b** and **c** are from UAV by Shoupeng Wang)



**2 Study area**

Northeast China is one of three main Mollisols regions in the world, with a total area of $7.0 \times 10^4$ km$^2$, contributing 20% of the grain and more than 40% of the maize in China. Most of the Mollisols region was gradually converted from native vegetation to cropland beginning in the late 19th Century. Cropland constitutes 80% of the total land area, and its main crop types are soybeans and corn. The study area is located in the typically heavy gully erosion area of the Mollisols region of Northeast China, where the native grasslands and forests have been completely converted into cropland since 1968. Additionally, it is situated in the transitional rolling hilly areas extending from the Songnen Plain to the Greater Khingan Mountains in the west, the Lesser Khingan Mountains in the north, and is near the Nen River (Fig.1a). Due to the gentle landscape, the farmland in the study area is covered by a thick black-organic soil layer, with sandstone, mudstone, and sandy conglomerate underneath.

The observed two permanent gullies in this work are 1.4 km apart and are located on south-facing and north-facing rolling-slope respectively (Figs. 1b and 1c). They are still expanding because they directly connect to the river network, which drains water to the Nen River. The width of gully No. I gradually broadens while that of gully No. II becomes narrow, and the depth of gully No. I is deeper (Figs. 2a and 2b). In detail, the gully depth of the No. I is 3.5m while that of gully No. II is 0.7m. Though the grass covers near the sidewall and the ridge along the gully, mass-wasting movement frequently occurs during the melting season and rainy seasons. The differences in gully planform and depth indicate that the mass movement at the sidewall or head cut have distinctive rates and scale. We have preliminarily found that the mass movement at the sidewall of the two gullies notably differs in scale, as exemplified by the Figs. 2c and 2d. Moreover, the height and width of gully No. II are lower than those of gully No. I (Fig. 3), and the head-cut area of gully No. I experienced tillage activities, while the headcut area of gully No. I does not. Therefore, the gully No. II can be deemed as the initial development stage for a large permanent gully.

The study area has a cold temperate continental monsoon climate with variable annual precipitations ranging from 480 mm to 512 mm, and 600 mm on average, according to the meteorological records at Nen Jiang City ( located 18.8 km from the study area). The rainfall mainly occurs between June and August, accounting for 70%–90% of the annual precipitation, with 461 mm on average. The snowfall occurs mainly from November through April, accounting for 10%–30% of the annual precipitation. The average temperature in the coldest and warmest months are –22.5 °C and 20.8 °C, respectively, with an annual average temperature of 0 °C.

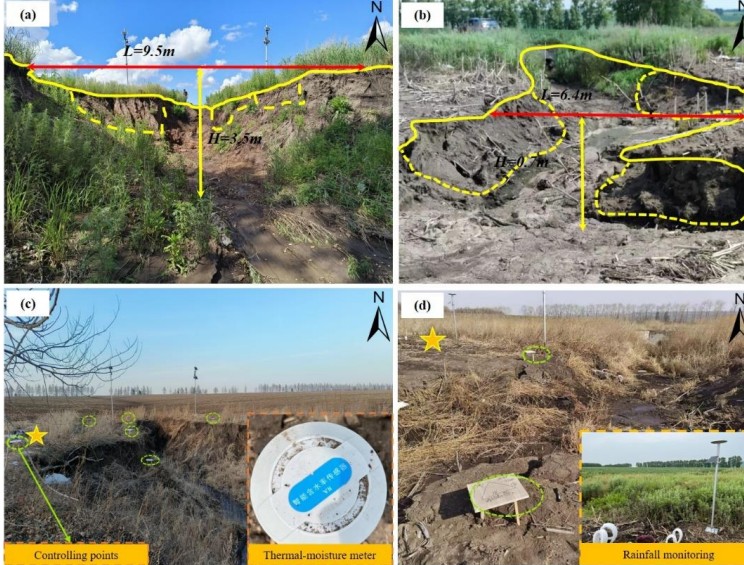

**Fig. 2.** A close view of the over-steepen slope and headcut of the two permanent gullies, with **(a)** cross section and
upstream view of the permanent gully No. I, **(b)** cross section and downstream view of the permanent gully No.
II, **(c)** ground controlling points (blue dot circles) and the soil moisture-temperature monitoring site (yellow
star) at permanent gully No. I, and **(d)** ground controlling points and the soil moisture-temperature monitoring
sites at permanent gully No. II.

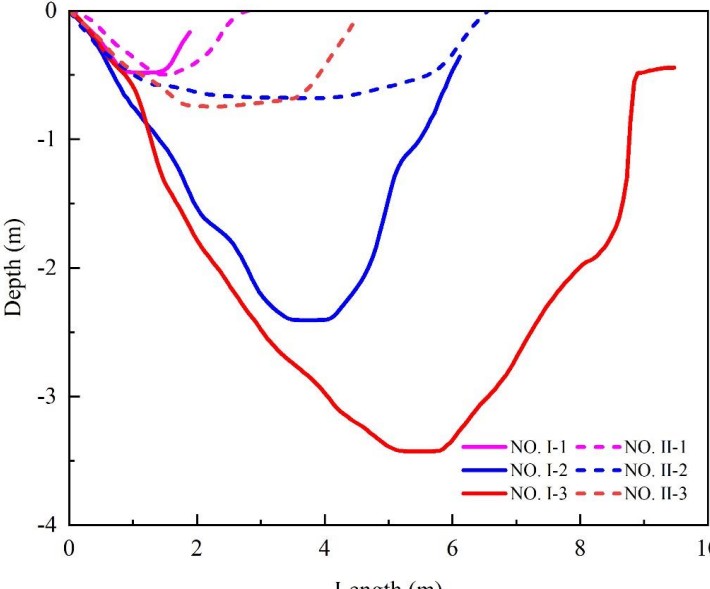


**Fig. 3.** Difference of the two permanent gullies' cross section. The location of the cross-section lines is shown in
figs.1b and 1c.
**3 Material and methods**
**3.1 Observation work**

Near the gully head cut, the hydro-thermal meter (frequency domain reflectometry sensor, FDR) was installed
to monitor the soil moisture and temperature at depths of 20cm, 40cm, 60cm, and 80 cm and air temperature (Fig.
2c). These two monitoring sites share the same rainfall records at gully No. II (Fig. 2d). Furthermore, a trench was
dug to obtain the soil samples at these two monitoring sites. The soil samples were used for particle component
analysis by the Malvern Mastersizer 3000 instrument (Malvern, UK), pore water pressure dissipation test by
consolidated undrained triaxial compression test (CU) using the GDS triaxial apparatus (GDS, UK), and unsaturated
permeability measurement using the transient release and imbibition method (TRIM; Lu and Godt, 2013). These two
methods focused on the hydraulic conductivity regarding the pore water release at varied confining stress and the
Soil Water Characteristic Curve (SWCC), and Hydraulic Conductivity Function (HCF), respectively.

To observe the gravitational mass-wasting process during the rainy and melting seasons, the study area was
scanned by numerous control plates (the dots in Figs. 1a and 1b, and dashed circles in Figs. 2c and 2d), installed in
and around the gully area and used unmanned aerial vehicle (UAV). These control points were used to analyze the
accuracy of the UAV-derived map and digital elevation model, aiming to obtain highly accurate topography. Then,
the differences between two digital elevation models generated the positive and negative terrain, which quantitatively
showed the erosion intensity of the gravitational mass-wasting. Additionally, the eroded soil volume in the unit over-



steepen slope surface area, termed areal erosion intensity, was applied in this work to address the erosion intensity
of gravitational mass-wasting.

**3.2 Pore water pressure rising and dissipation**

The soil samples were initially saturated in a vacuum pump, followed by consolidated in the chamber of a GDS
apparatus by 100, 200, and 300 kPa effective confining pressure with a 10-kPa backpressure. The consolidating
process completed when the pore water pressure decreased to the values of backpressure.
For pore water increasing stage:

$$P_\uparrow = P_0 t^{b_\uparrow} \tag{1}$$

where $P_\uparrow$ is the measured pore water pressure during the increasing stage (kPa), $P_0$ is the initial pore water pressure
since loading (kPa), $t$ is the time (s), $b_\uparrow$ is the rising proxy reflecting the steepness of the power-law curves of pore
water pressure increase.
For pore water dissipation stage:

$$P_\downarrow = \frac{P_{max}}{1 + b_\downarrow t} \tag{2}$$

where $P_\downarrow$ is the measured pore water pressure during the dissipation stage (kPa), $P_{max}$ is the maximal pore water
pressure since loading (kPa) and is the rollover point in the pore water pressure curve, $t$ is the time (s), $b_\downarrow$ is the
dissipation proxy reflecting the water drainage ability of soil mass at given confining pressure.
reflects the concavity of the pore water pressure dissipation curve.

**3.3 Hydro-mechanical property**

The transient Release and Imbibition method (TRIM) was performed to test the unsaturated permeability of
soil mass (Lu and Godt, 2013). The Soil Water Characteristic Curve (SWCC) and Hydraulic Conductivity Function
(HCF) were obtained using Hydrus 1-D (Wayllace and Lu, 2012). Adopting the models proposed by Mualem (1976)
and van Genuchten (1980), the constitutive relations between the suction head ($h$), water content ($\theta$), and hydraulic
conductivity ($K$) under drying and wetting states can be represented by the following equation:

$$\frac{\theta - \theta_r}{\theta_s - \theta_r} = \left[\frac{1}{1 + (\alpha|h|)^n}\right]^{1 - \frac{1}{n}} \tag{3}$$

and

$$K = K_s \frac{\left\{1 - (\alpha|h|)^{n-1}[1 + (\alpha|h|)^n]^{\frac{1}{n} - 1}\right\}^2}{[1 + (\alpha|h|)^n]^{\frac{1}{2} - \frac{1}{2n}}} \tag{4}$$

where $\theta_r$ is the residual moisture content (%), $\theta_s$ is the saturated moisture content (%), $\alpha$ and $n$ are empirical
fitting parameters, $\alpha$ is the inverse of the air-entry pressure head, $n$ is the pore size distribution parameter, and $K_s$
is the saturated hydraulic conductivity (cm/s).
Based on the observed volumetric water content and the SWCC, the suction stress ($\sigma^s$, kPa) throughout the
observation stage can be expressed as:

$$\sigma^s = -\frac{S_e}{\alpha}\left(S_e^{n/(1-n)} - 1\right)^{1/n} \tag{5}$$

**3.4 Water storage and leakage**

In this study, the hydrology process of the overs-steepen slope is of utmost importance for analyzing the
gravitational mass-wasting because of the varied soil water storage and drainage in rainy and snow-melting seasons.
In fact, soil water is stored during rainstorms, while it drains after the rainstorm cease. The drainage process during
the melting process will not be addressed herein because the melting water constantly contributes to high soil
moisture. Therefore, the soil water storage ($S_s$) during rainstorms and the snow-melting season, and drainage ($S_d$)
after rainstorms cease can be evaluated by the soil depth and the difference between the maximum soil moisture and
antecedent soil moisture:



$$S_e = \frac{\theta - \theta_r}{\theta_s - \theta_r} \tag{6}$$
$$S_s = S_e^w \Delta h_i \tag{7}$$
$$S_d = P - S_e^d \Delta h \tag{8}$$
where $S_e$ is the degree of saturation, $\theta$ is the in-situ observed volumetric moisture content measured (%), $\Delta h_i$ is
the soil layer $i$ (200 mm in this work, $i$=1 , 2, 3, 4), $S_e^w$ and $S_e^d$ are the residual soil moisture in the wetting and
drying processes (%), and $P$ is the accumulated rainfall (mm) and equals to 0 mm in snow-melting season.

## 4. Results


### 4.1 Areal erosion intensity of gully bed and over-steepen slope


During the melting season in 2023 and the rainy season in 2022, three high-resolution maps and digital elevation
models (DEM) of the two permanent gullies on 28 June 2022, 17 October 2022, and 20 June 2023, were obtained
with high resolutions of 0.058, 0.108 and 0.042m, respectively. The DEMs were spatially registered in ArcGIS 10.2
by a standard layer of orthoimage, ground control points, and the spline function. Then, the erosion depth of the
headcut area could be obtained by the differences between two DEMs. Therefore, the linear and the areal erosion
intensity can be calculated using the erosion depth and the grid size (Fig. 4).

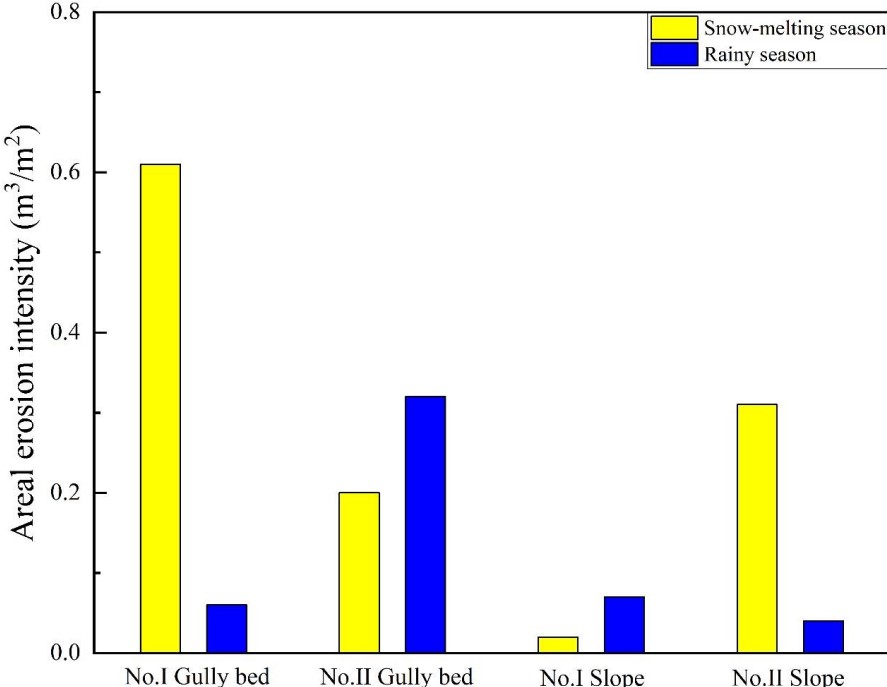


**Fig. 4.** Differences in the areal erosion intensity for gully bed and over-steepen slope
The undercutting of the channel bed mainly resulted from the sediment delivery by channelized water flow.
The areal erosion intensity in the snow-melting season for gully No. I was greater than that in gully No. II, which
could be driven by the low melting water storage and high melting water runoff at the headcut of gully No. I. In the
rainy season, the areal erosion intensity of gully No. II was notably greater than that in gully No. I, which may result
from the rapid water storage and leakage, producing intensive runoff at the headcut of gully No. II. The erosion of
the over-steepen slope was mainly from the gravitational mass-wasting process. For gully No. II, the areal erosion



intensity in the snow-melting season was significantly greater compared to that in the rainy season. In the snow-
melting season, the areal erosion intensity for gully No. II was greater than that in gully No. I. Though the areal
erosion intensity in the rainy season for gully No. I was higher than that for gully No. II, the difference was as
negligible as that in snow-melting season. It is important to note that the slopes of the permanent gully were over-
steepen, and the stability of the slope primarily depended on the soil suction stress, as a function of the hydro-
mechanical properties and the soil moisture.
In the study area, Tang et al (2023) addressed the eroded volume in permanent gully with drainage area and
rainfall, while the erosion intensity of the channel bed and the over-steepen slope and their main influencing factors
were not documented. As the channel bed erosion was closely correlated with the hydrological process, and the over-
steepen slope erosion corresponded to the soil suction stress status in unsaturated conditions, it requires further
examination of the associated soil water storage and leakage, as well as the hydro-mechanical properties of the soil
mass in the two permanent gullies. Importantly, one of the differences in the hydrology process at the headcut area
indicates that both soil water storage and leakage exist in the rainy season. Moreover, the water leakage process was
absent during the snow-melting season. This results from continuous melting water from surfical snow and ice in
the macro-pores and fissures. Once the melting process was completed, the soil water storage process ceased with
the onset of water leakage process during the transition time between the snow-melting season and the rainy season.

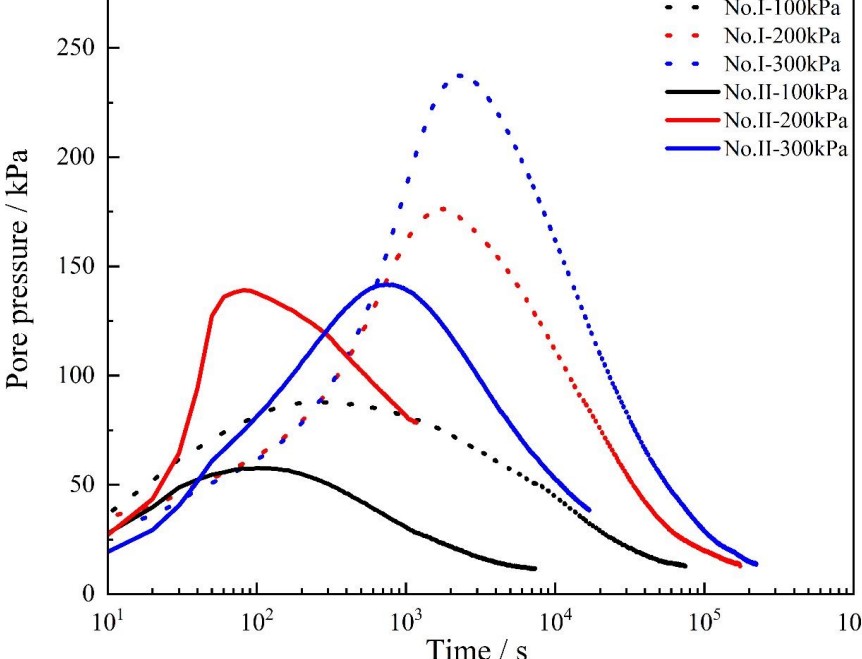


**Fig. 5.** Variation in pore water pressure under effective confining pressure of 100, 200 and 200 kPa by GDS triaxial
shear tests (GDS Instruments, UK). The proxy for the pore water pressure rising and dissipation are calculated
by Eqs. (1) and (2). The rising and dissipation ratio are calculated by pore water pressure difference during
given time interval. The values of proxy and ratio are shown in table 1.
**4.2 Physical properties of Mollisols**
**4.2.1 Pore water pressure rising and dissipation**





The consolidation module of the GDS apparatus was used to measure the pore water pressure generation and
dissipation, by consolidating soil and draining water from the initial saturated state. Under the same confining
pressure, pronounced differences were observed in the rising and dissipation ratio of the pore water pressure within
the Mollisols of the two gullies. The results of the pore water pressure during the consolidation process under 100,
200, and 300 kPa effective confining pressure were compared (Fig. 5). Meanwhile, the physical properties as well
as the rising and dissipation-ratio and proxy were shown in Table 1.
**Table 1.** The physical properties and pore water pressure changes of the soil mass

| Parameters | Definition | Confining pressure (kPa) | Permanent gully | |
| --- | --- | --- | --- | --- |
| | | | No. I | No. II |
| $v_\uparrow$ (kPa/min) | Pore water rising ratio | 100 | 11.83 | 23.04 |
| | | 200 | 4.86 | 90.52 |
| | | 300 | 5.55 | 10.92 |
| $b_\uparrow$ | Pore water rising proxy as eq. (1) | 100 | 0.23 | 0.25 |
| | | 200 | 0.24 | 0.46 |
| | | 300 | 0.30 | 0.41 |
| $v_\downarrow$ (kPa/h) | Pore water dissipation ratio | 100 | 3.68 | 22.77 |
| | | 200 | 3.32 | 194.47 |
| | | 300 | 3.66 | 23.94 |
| $b_\downarrow$ ($\times 10^{-5}$) | Pore water dissipation proxy as eq. (2) | 100 | 9.97 | 79.70 |
| | | 200 | 7.80 | 79.40 |
| | | 300 | 6.82 | 18.10 |
| $c$ (kPa) | Effective cohesion | | 11.3 | 7.2 |
| $\varphi$ (°) | Effective friction angle | | 16.3 | 21.3 |
| $\gamma$ (kN m$^{-3}$) | Unit weight | | 14.1 | 12.5 |

Overall, the peak value of pore water pressure within the Mollisols of gully No. I was notably higher than that
of gully No. II. Moreover, the peak value of the pore water pressure within the Mollisols of gully No. II increased
to 57.6, 139.0, and 141.7 kPa under confining stress of 100, 200, and 300 kPa, respectively. In contrast, the peak
value of the pore water pressure within the Mollisols of gully No. I increased to 87.9, 176.1, and 237.3 kPa,
respectively. The high peak pore water pressure sufficiently illustrates that the Mollisols of gully No. II had strong
hydraulic conductivity as the increasing ratio and proxy and dissipation ratio and proxy represent the pore
connectivity. During the rising stage, the rising ratio for the Mollisols of gully No. II was 2 to 18.6 times greater,
and its rising proxy was 1.08 to 1.92 times larger than those of gully No. I. Within the dissipation stage, the ratios
were 6.20 to 58.6 greater, and its proxies were 2.65 to 8.0 times larger than those for Mollisols of gully No. I.
Additionally, the largest difference between these two gullies was obtained under the confining stress of 200 kPa.
Therefore, the pore water pressure rising and dissipation properties suggest that the headcut area of gully No. II may
exhibit an active hydrology process.
**4.2.2 Hydro-mechanical property**
Fig. 6 shows the results of the TRIM tests, soil water characteristic curve (SWCC), hydraulic conductivity
function (HCF), and estimated suction stress with varied saturation degrees. The water outflow mass was measured
at 10-min intervals during the drying and wetting processes. The water outflow masses measured for Mollisols of
gully No. II were generally higher than those for the Mollisols of gully No. I. For the drying tests using the Mollisols
of gully No. II and No. I, the given water outflow masses were 0.0713 and 0.060 g per 10 min, respectively.
Meanwhile, for the wetting tests, the given water outflow masses were 0.031 and 0.0208 g per 10 min, respectively





(Fig. 6a). Overall, the permeability of Mollisols gully No. II was higher than that in the Mollisols gully No. I. The
same results were found in the pore water pressure rising and dissipation ratio and proxy in Table 1.

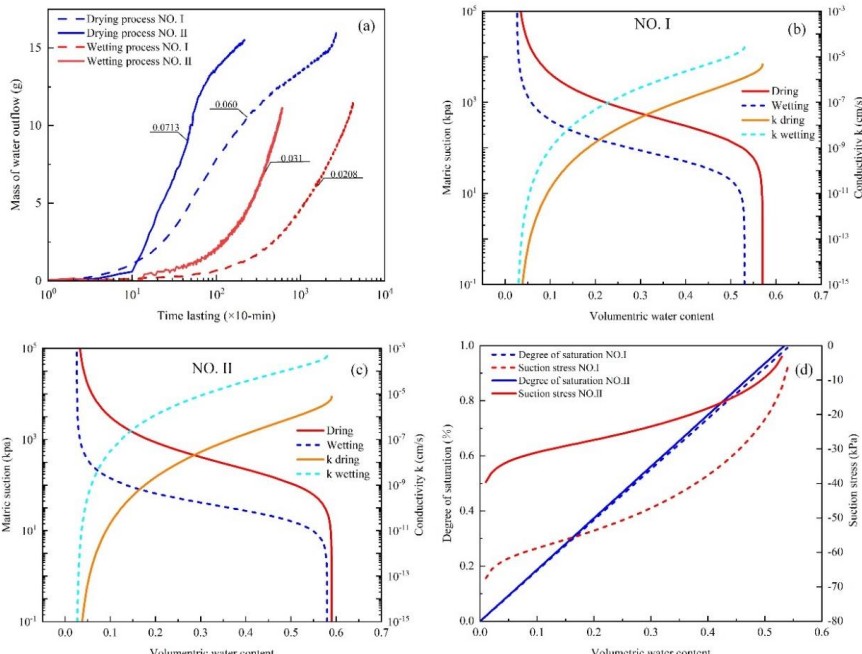


**Fig. 6.** Differences of the hydro-mechanical properties of the two soil masses. **(a)** Water flow mass in drying and

wetting process. **(b)** Soil water characteristics curve for soil mass of permanent gully No. I. **(c)** Soi water

characteristics curve for soil mass of permanent gully No. II. **(d)** Suction stress-volumetric water content curves

for the two soil masses. The mass of water outflow was recorded at 10 min for each test.

**Table 2.** Parameters describing the soil and water characteristic curve (SWCC) and the hydraulic conductivity
function (HCF) from Hydrus 1D.

| Parameters | Definition | Permanent gully | |
|---|---|---|---|
| | | NO. I | NO. II |
| $\theta_r$ | Residual moisture | 0.0262 | 0.0259 |
| $\theta_s^d$ | Saturated moisture | 0.57 | 0.59 |
| $\theta_s^w$ | | 0.53 | 0.58 |
| $\alpha^d$ (kPa$^{-1}$) | The inverse of the air-entry pressure head | 0.0042 | 0.0063 |
| $\alpha^w$ (kPa$^{-1}$) | | 0.0183 | 0.0375 |
| $n^d$ | The pore size distribution parameter | 1.69 | 1.68 |
| $n^w$ | | 1.95 | 1.91 |
| $K_s^d$ (cm s$^{-1}$) | Saturated hydraulic conductivity | 4.73×10$^{-6}$ | 7.82×10$^{-6}$ |
| $K_s^w$ (cm s$^{-1}$) | | 2.64×10$^{-5}$ | 4.26×10$^{-4}$ |

Notes: the superscript $d$ and $w$ indicate drying and wetting states.



As the observation stage in this study covered the rainy and snow-melting seasons, the tensiometer was not adopted to monitor the matrix suction due to the potential damage to the high-air-entry (HAE) ceramic plate under either strong freezing or active drying-wetting processes. Therefore, the soil water characteristic curve (SWCC) was chosen to observe soil moisture for estimating the suction stress. The SWCC of Mollisols in the two permanent gullies were obtained using the Hydrus-1D code with the reverse modeling option, and the Levenberg–Marquardt nonlinear optimization algorithm (Lu and Godt, 2013). Table 2 shows the soil parameters obtained using the Hydrus 1D inversion.

Using the parameters in Table 2, the SWCC and HCF curves of the Mollisols were plotted (Figs. 6b and 6c). Air entry pressure and residual water content are two important parameters that describe the hydrological and mechanical characteristics of Mollisols. The air entry pressure represents the critical value at which air enters the saturated soil and starts to drain. In comparison, the values of $\alpha^d$ and $\alpha^d$ together prove that the required air entry pressure for Mollisols in gully No. I is greater than that in gully No. II. and the differences were 79.4 kPa and 28.0 kPa under drying and wetting conditions, respectively (Table 2). Therefore, water infiltration in gully No. II during either the rainy or snow-melting season was more active compared to gully No. I. The residual moisture does not vary significantly due to the similarity in soil types.

The saturated hydraulic conductivities of Mollisols in gully No. I was lower than those in gully No. II in both the drying and wetting processes. In Table 1 and Fig. 5, the pore water pressure rising ratio and proxy, as well as the dissipation ratio and proxy, further indicate that the permeability of Mollisols in gully No. II was greater than that in Mollisols in gully No. I. Therefore, pore water pressure changed in varied confining stress, air entry pressure and the saturated hydraulic conductivities under drying and wetting conditions suggest that it is more challenging for the Mollisols in gully No. I to absorb and drain water compared to the Mollisols in gully No. II.

Figs. 6b and 6c show the matrix suction and hydraulic conductivity at varied soil moisture. However, it was unable to compare the level of suction stress with varied hydrological-mechanical parameters as shown in Table 1. Hence, the level of suction stress at varied soil moisture was provided (Fig. 6d). The absolute of the suction stress at specified soil moisture for Mollisols in gully No. I was higher than that for Mollisols in gully No. II, suggesting a higher possibility of gravitational mass-wasting occurrence for Mollisols in gully No. II.

**4.3 Hydrological response**

**4.3.1 Monitoring results**

To display the water storage during the rainy and snow-melting seasons, as well as the water drainage after the rainfall, comprehensive information was considered, including rainfall amount, air temperature, soil moisture and temperature at various soil layers (Fig. 7). The recorded rain events were categorized into four groups: light rain, moderate rain, torrential rain and rainstorms, with rain amounts of < 10 mm, 10−25 mm, 25−25 mm, and 50−100 mm, respectively (Fig. 7a). In total, 24 light rain events, 2 moderate rain events, 5 torrential rain events and 1 rainstorm were recorded. During the snow-melting season, the air temperature started to increase above 0°C on 20 March with an increasing gradient of 0.15°C, which reached 2.3°C after 23 April (Fig. 7b). For the soil moisture changes, the volumetric water content for the 20 cm depth for gully No. II had greatly increased since 23 April, while it only slightly increased for gully No. I. This suggests that the headcut area of gully No. II may be experiencing a high soil moisture level.

The soil moisture levels throughout the rainy and snow-melting seasons displayed notable dissimilarities between these two sites. During the rainy season, the volumetric water content at a depth of 20 cm remained persistently at a lower level of soil moisture than the other three soil depths, as shown in Fig. 7c. However, during the snow-melting season, the volumetric water content for 40 cm soil layer was the largest (Fig. 7d). Overall, the soil moisture of gully No. II, either in the rainy or snow-melting season, exhibited greater fluctuations compared to gully No. I. These results indicate that water infiltration from rain events or snow-melting water for the headcut of



gully No. II was more active compared to gully No. I. Furthermore, the observed difference proves that the stored
and leaked water for the headcut of gully No. II was significantly greater than that of gully No. I.

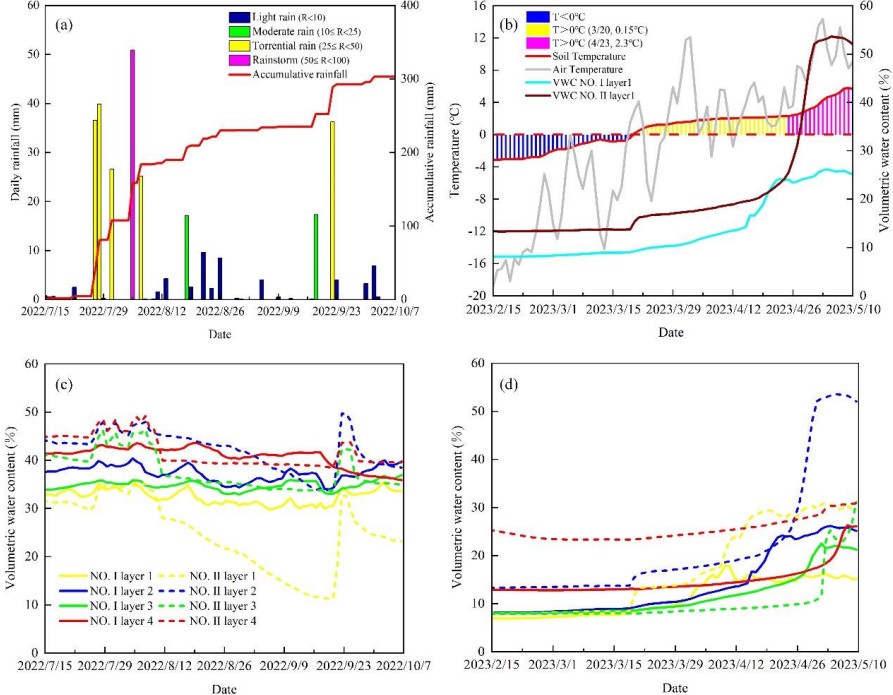

**Fig. 7.** Field-monitored rainfall conditions, air and ground temperate, and volumetric water content. **(a)** Rain pattern
during the rainy season. **(b)** Soil, air temperature and surficial volumetric water content during the snow-melting
season. **(c)** and **(d)** Monitored volumetric water content during the rainy season and snow-melting season.

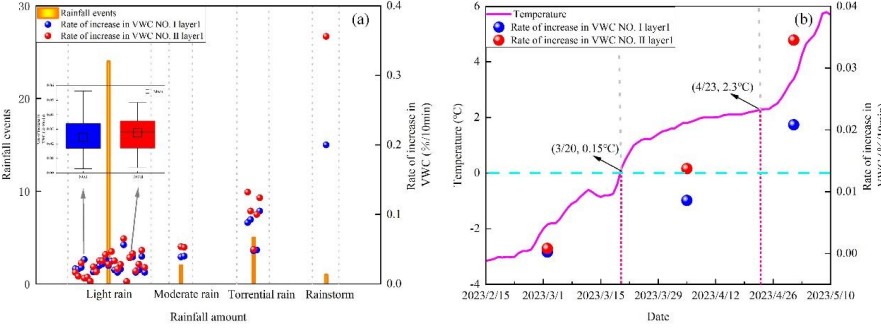

**Fig. 8.** Volumetric water content increasing ratio in snow-melting ratio and the rainy season. (a) Rate of increasing
of VWC at varied rain events. **(b)** Rate of increasing of VWC at three temperature increasing stage.

To further analyze the difference in water infiltration during the rainy and snow-melting seasons, the rate of
soil moisture increased at a depth of 20 cm was compared in detail (Fig. 8). In the four types of rain events, the mean
increasing rates for gully No. II were 0.027, 0.053, 0.102, and 0.356, respectively, which were 1.12, 1.35, 1.34, and
1.78 larger than those for gully No. I (Fig. 8a). During the snow-melting season, the soil moisture increasing ratio in
the initial, medium, and final stages for the gully No. II were 3.48, 1.60, and 1.66 times those in gully No. I (Fig.



8b). Therefore, the water infiltrate ratios for the headcut area of gully No. II during the rainy and snow-melting
seasons were greater.

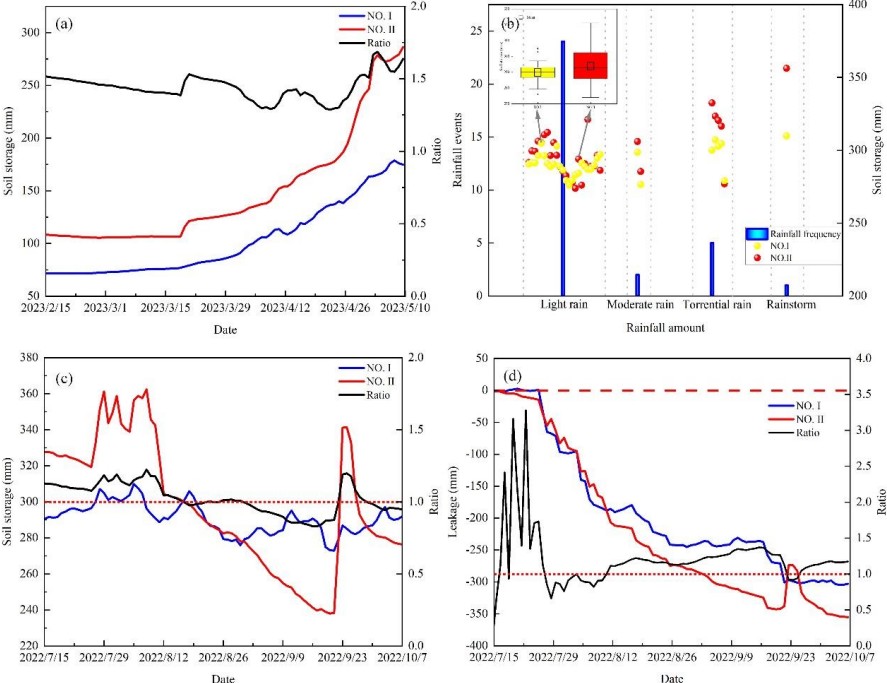


**Fig. 9.** Hydrological response during the rainy and snow-melting season. **(a)** Soil water storage and the storage ratio
during the snow-melting season. **(b)** Soil water storage at varied rain events. **(c)** Soil water storage and the
storage ratio for the two permanent gullies. **(d)** Soil water leakage and the leakage ratio during the rainy season.
During the rainy season, water storage and leakage synchronously change with the onset and end of rainfall.
**4.3.2 Water storage and leakage**
The water storage and leakage for the headcut of the permanent gully are closely related to the erosion intensity.
In general, intensive storms in erosive rain events generate strong erosion due to intensified water flow. For the
events with rainfall amounts exceeding the erosive rain events, high water storage decreased the surface runoff and
led to fewer erosions (Tange et al., 2023). Furthermore, the response of the water storage and the leakage is closely
related to rainfall amounts and temperatures. Rapid water storage and leakage process corresponds to the
permeability of the soil mass.
Fig. 9 shows that the stored and leaked water for the soil column at the headcut of the two gullies. In the snow-
melting season, the stored water in gully No. II was greater compared to gully No. I. The stored water ratio was
calculated by dividing the stored water amount of gully No. II by the amount stored in gully No. I, was typically
larger than 1.0 throughout the snow-melting season (Fig. 9a). In particular, the ratio increased abruptly since 26
April. Therefore, the stored water amount for the headcut of gully No. II was higher.
Regarding four types of rain events, the mean stored water for the headcut of gully No. II during 24 light rain
events was greater compared to gully No. I (Fig. 9b). Specifically, the differences in stored water at the headcut of
the two gullies were 4.0, 8.1, 15.2, and 46.3 mm, respectively. Therefore, the stored water, either in the snow-melting
season or rainy season, was higher in the headcut of gully No. II. However, the stored water at the headcut of gully
No. II was not always higher than that of gully No. I, as exemplified by Fig. 9c. During 26 August and 3 September





2022, the stored water at the headcut of gully No. II was less than that of gully No. I, which could be attributed to
the high temperature and light rain events. However, the stored water at the headcut of gully No. II exceeded that of
gully No. I during the torrential rain event on 22 September. Furthermore, the water storage for gully No. II exhibited
stronger fluctuations. In general, rapid water infiltration coincided with instant water leakage. Fig. 9d shows the
water leakage and the leakage ratio of the two gullies during the rainy season. The leaked water for gully No. II was
higher than that of gully No. I. Therefore, the headcut area of gully No. II performed a better water storage ability
in snow-melting and rainy seasons and more rapid water leakage in rainy season than that of gully No. I.
In summary, rapid water storage and leakage for the headcut of gully No. II during the torrential rain or
rainstorm coincided with both observed pore water pressure rising and dissipation, as well as the hydro-mechanical
properties of Mollisols. The high permeability of Mollisols at the headcut of gully No. II was attributed to the more
rapid water storage, leakage process, and stored water. This consequence could have a considerable influence on the
erosion tensity on the over-steepen slope and gully bed of the permanent gully.

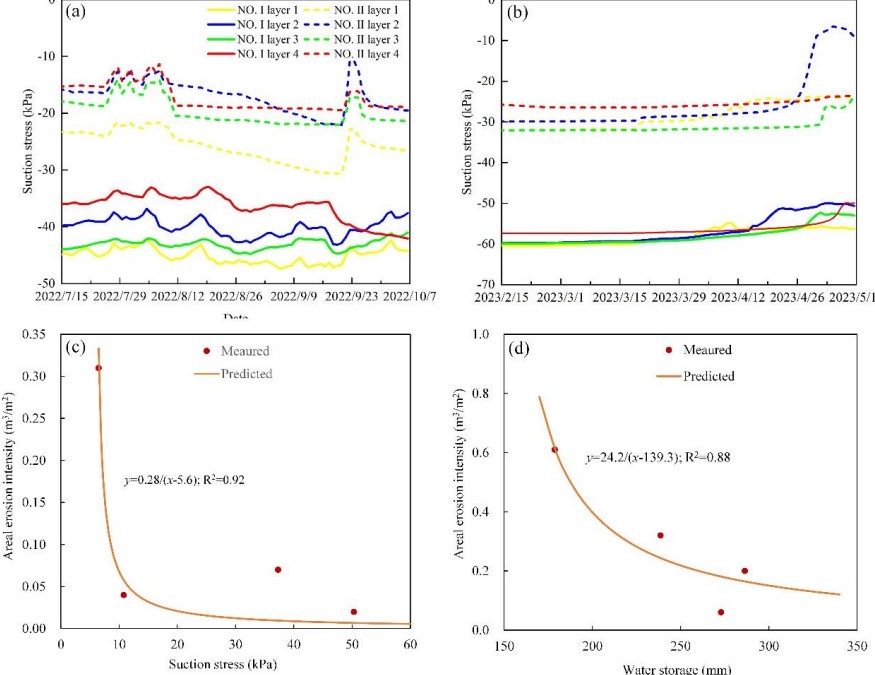


**Fig. 10.** Relationship between hydrology and the hydro-mechanical state with the erosion intensity. **(a)** Suction stress
during the rainy season. **(b)** Suction stress during the snow-melting season. **(c)** Areal erosion intensity on over-
steepen slope decreases with suction stress. **(d)** Areal erosion intensity on channel bed decreases with water
storage amount.
**4.4 Hydro-mechanical response and erosion intensity**
As the Mollisols at the headcut area of two permanent gullies differed in hydro-mechanical properties, the soil
moisture monitored varied greatly in the field. The suction stress was estimated according to the field-monitored soil
moisture at each site and the relationship between soil moisture and matrix suction (Figs. 6d, 7c, and 7d). During the
rainy season, the absolute values of suction stress of Mollisols for gully No. II were less compared to gully No. I
(Fig. 10a). The smaller absolute values of suction stress for Mollisols gully No. II during the snow-melting season
illustrated in Fig. 10b. Moreover, the smaller suction stress in the snow-melting season may result in a strong erosion





in the over-steepen slope of gully No. II, as exemplified by Fig. 4.
Because the hydrology process of the headcut area was closely related to the channel bed erosion, the hydro-
mechanical response influenced the stability of the over-steepen slope. It is important to analyze the possible
relationship between the areal erosion intensity on the channel bed, water storage, and erosion intensity of the over-
steepen slope with suction stress. In general, the high absolute value of suction stress aligned with strong cohesive
forces between soil particles, which is a sign of stability. In contrast, a low absolute value of suction stress suggests
a higher possibility of slope failures. Therefore, the relationship between the absolute value of suction stress and
areal intensity could be negative. Fig. 10a exhibits the reciprocal relationship between the suction stress and areal
erosion intensity of over-steepen slope. The empirical relationship indicates that gravitational mass-wasting on the
over-steepen slope occurred and the permanent gully expanded when the suction stress remained at a relatively low
suction stress in a prolonged period, particularly about 5.6 kPa for the study area.
Erosion on the channel bed was closely related to the runoff discharge during erosive rain events. In fact,
numerous studies have examined the soil loss during gully erosion. In this study area, the contributing area and the
rainfall amount of erosive rain events were used to develop the soil loss estimation equation (Tang et al., 2023).
During erosive rain events, the amount of water storage decreased the runoff amount and intensity. In other words,
the less rainwater was stored during erosive rain events, the higher the runoff amount or the more intensive the
channeled flow. Therefore, the relationship between water storage and the areal erosion intensity of the channel bed
could be negative. Fig. 10d shows the reciprocal relationship between the areal erosion intensity of the channel bed
and water storage. It indicates that excessive rainwater in erosive rain events could create intensified channeled flow
to erode the channel bed if the stored water in the Mollisols reached a threshold, such as 139.3 mm in this study area.
The rainfall amount of 139.3 mm in this study was smaller than 177 mm proposed by Tang et al (2023), which may
result from plant interception and depression detention during the rainy season.
**5 Discussions**
The physical process of permanent gully development can be categorized into gravitational mass-wasting
process on the over-steepen slope and sediment delivery on the channel bed (Montgomery and Dietrich, 1992; van
Beek et al., 2008; Luffman et al., 2015). Traditionally, the majority of studies on the gully erosion have focused on
the soil loss from water erosion, and soil loss estimation typically was based on upslope contributing area,
topographic conditions, erosive rainfall factors, and land use conditions (Li et al., 2015; Xu et al., 2017; Wang et al.,
2021; Tang et al., 2022). It is worth noting that the physical mechanics of bed erosion and slope erosion are notably
different, making it challenging to accurately predict soil loss on the over-steepen slope in the permanent gully.
Additionally, the gravitational mass-wasting process on the over-steepen slope differs slightly from the rainfall-
induced shallow landslides especially for those without given failure planes (Poesen et al., 1998; Guo et al., 2020),
although they share similarities such as the decrease of soil strength due to water infiltration (Guo et al., 2019). Thus,
a thorough mechanics analysis is necessary to comprehend the physical process of gravitational mass-wasting
process on the over-steepen slope and sediment delivery on the channel bed.
This study thoroughly investigated the effects of hydrological factors and hydro-mechanical properties of soil
mass on the erosion intensity on the over-steepen slope and channel bed because the mass failure on hillslopes was
closely related to the status of suction stress in unsaturated conditions, and the erosion on channel beds was notably
influenced by water storage or runoff amount. Therefore, the hydrological factors referred to water storage and
leakage (Fig. 9) and volumetric increasing ratio at various rain events and snow-melting stages (Fig. 8). The hydro-
mechanical properties included the pore water pressure rising and dissipation (Table 1 and Fig. 5), the saturation
degree-suction stress of soil mass (Fig. 6), and estimated suction stress during rainy and snow-melting seasons (Figs.
10a and 10b). Importantly, two permanent gullies were selected on the field observations that the erosion patterns of



the over-steepen slope and the gully bed varied discernably. The headcut area of gully No. II showed signs of
disruption, in contrast to gully No. I., resulting in disparities in the areal erosion intensity, either by season or site.
The hydro-mechanical properties of Mollisols in the two gullies clearly exhibit the capability of the water movement
under given confining stress, as exemplified by the pore water pressure rising and dissipation ratio and proxy. At the
headcut of gully No. II, the Mollisols were greatly disturbed, and the soil mass had higher permeability and lower
suction stress at a specified saturate degree. This finding indicates that water infiltration, from either rain events or
the snow-melting, was more active at the headcut area of gully No. II compared to gully No. I, as evidenced by water
storage and leakage, as well as volumetric water content increasing ratio. Importantly, the ratio of the volumetric
water content increasement increases as the rain group and the temperature. Therefore, the headcut area of gully No.
II experienced more aggressive hydrology processes.

The main contribution of this work is the analysis of areal erosion intensity through hydrology and hydro-

mechanical responses. The erosion of the channel bed was related to the runoff amount or intensity. However, this
work did not address the runoff issue, focusing instead on water storage. In fact, water storage and the runoff depth
were approximately equal to the rainfall depth. Therefore, the areal erosion intensity of the channel bed was inversely
proportional to the water storage, as exemplified by Fig. 10d. Regarding erosion on the over-steepen slope, some
scholars have summarized that the combination of a long-duration storm (Xu et al., 2020), initial soil moisture in the
pre-winter season (Wen et al., 2024), tensile crack morphology (Zhou et al., 2024) and heaving and thawing (Thomas
et al., 2009) could trigger mass failures on the bank. However, it is still unclear how the stability of the over-steep
slope responded to the soil moisture. Long-term saturation could provide sufficient water infiltration and low suction
stress. The highest areal erosion intensity occurred in the snow-melting season, not in the rainy season (Fig. 10c), as
the duration of snow-melting was longer than that of rain events (Figs. 7a and 7b). In this study area, Dong et al
(2011) previously revealed that the critical mass water content for gravitational mass-wasting ranged from 31.0% to
33.8%. This mass water content range corresponded to a volumetric water content of 39.0% to 48.0% for the soil
mass, with a suction stress of 11.0 kPa. The differences revealed the limitation of the direct-shear apparatus, which
could only provide total cohesion without the ability to separate the contribution of effective cohesion and the suction
stress on the total cohesion. As noted by Xu et al (2020) and shown in Fig. 10b, the high water storage during the
snow-melting season in gully No. II (Fig. 9a) and long-term water infiltration can result in low suction stress and
high areal erosion intensity of the over-steepen slope. Thus, the relationship between the absolute suction stress and
areal erosion intensity can be reciprocal. Lastly, the accuracy of two empirical equations, as shown in Fig. 10, could
be improved in the future if additional monitor sites were added or if the study period were extended to cover multiple
rainy-snow seasons.

## 6 Conclusions

Permanent gully development is considered a hydrogeomorphic phenomenon and its physical mechanics can

be attributed to the hydrology and hydro-mechanical responses of the headcut. In the Mollisols region of Northeast
China, tremendous studies on gully development have focused on soil loss in response to rainfall or snow depth, but
limited documents have addressed the physical mechanics of gravitational mass-wasting. This study provides a
comprehensive analysis of erosion intensity on over-steepen slopes and channel beds in two permanent gullies
according to hydrology processes such as the infiltration, storage, and leakage of soil water, as well as the hydro-
mechanical response such as changes in suction stress levels. The following conclusions were drawn:

(1) In comparison, the Mollisols at the headcut area of gully No. II exhibited higher permeability than those of

gully No. I, which could be attributed to the elevated ratio and proxy of pore water pressure rising and dissipation.
The TRIM test results prove that the saturated Mollisols of gully No. II drainage water more rapidly compared to
gully No. I due to high air-entry pressure and the saturated hydraulic conductivity during wetting and drying



conditions.
(2) The headcut area of gully No. II had stronger hydrological processes, characterized by the higher ratio of
soil moisture increase for the four types of rain events and the three snow-melting stages compared to gully No. I.
Soil water storage of gully No. II experienced greater fluctuations in torrential rain and rainstorm events. Overall,
the absolute suction of gully No. II was lower than that of the gully No. I, potentially triggering greater erosion on
the over-steepen slope.
(3) The relationships between erosion intensity on the over-steepen slope and the channel bed were analyzed
regarding the suction stress and water storage. Findings indicate that low suction stress and high soil water storage
could contribute to great gravitational mass-wasting and reduce channel bed erosion. These two empirical
relationships and their associated efficiencies could be improved through ongoing monitoring efforts to enhance the
accuracy of soil loss prediction in the future.
**Acknowledgements**
All authors declare that no conflict of interest exists. This work was study was supported by the National Key
Research and Development Program (Grant No. 2021YFD1500700). The authors extend their gratitude to the
colleges at the Jiusan Soil and Water Conservation Experimental Station, Beijing Normal University, for their help
during field investigations.
**Code/Data availability**
Any readers can contact Prof. Chao Ma as the corresponding author is willing to share the raw/processed data.
**Author contributions**
Prof. Ma launched this work based on his skills in gravitational mass-wasting and unsaturated soil mechanics, and
proposed the idea-ology of hydrology and hydro-mechanical condition in analyzing the gravitational mass-wasting.
Under the guidance of Prof. Ma, Mr. Dongshuo Zheng and Shoupeng Wang finished indoor tests of soil strength and
hydraulic-mechanical properties. Prof. Zhang helped determine the field observation sites. Dr. Dong improved the
manuscript's language proficiency. Dr. Jie Tang and Yanru Wen provided the research progress about the gravitational
mass-wasting on gully expansion in the study area.
**Competing interests**
All authors have declared that there were no conflicts of interests and competing interests.

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
