# Peer review of "Understanding the soil loss at the permanent gully headcut area in the Mollisols region of Northeast China 3 Chao Ma1, Shoupeng Wang1, Dongshuo Zheng1, Yan Zhang1, Jie Tang2, Yanru Wen3, Jie Dong4 4 1. School of So"

_EGUsphere, 2024_

## Author Comment (AC1)

This study observed erosion processes and some soil-hydrological properties at the headcut areas of two permanent gullies in Northeast China's Mollisols region during rainy and snow-melting seasons. Key parameters like soil moisture, temperature, and precipitation were investigated to understand water storage capacity, leakage processes, and suction stress levels. Although only two headcuts were monitored, I think the results could be of interest for the scientific community, but I would only recommend the publication of the work if a series of changes and improvements are carried out. In my opinion, the authors wrongly include methodological content and interpretations in the results section. These contents should be rightly placed in the corresponding section. Additionally, I miss some methodological details that I consider relevant (UAV data, processing, etc. see my comments below). Finally, figures must be notably improved, particularly the font size, please see my comments below. I detail these suggestions in the following lines:

Replies:

Thanks for your comments to improve the quality of this manuscript.

Firstly, the physical process of gravitational soil erosion in permanent gully has been long neglected in the soil-water conservation research field, while it plays an important role in permanent gully expansion and development. In this work, we clearly addressed their differences and similarities with the landslides and used the theory of unsaturated soil mechanics to study their physical process.

As you suggested, we should clearly state some methodological content and interpretations in the method part, not in the results part. Meanwhile, some short paragraph in the results part should be moved into discussion part. We also found that the texts in some figures are not clear enough to read and some errors exists. Therefore, we made a thorough revision of this manuscript and the figures.

We made a point-to-point response to your comments, as follows:

Comment 1

Abstract: I have doubts about the last sentence as the USLE was not designed to predict gully erosion. I would recommend to delete this sentence.

Reply 1

Done. We revised the last sentence of in the abstract. As you know, USLE is an empirically-based function. If a physically-based function is developed to predict the gully erosion, it will be better. We revised the last sentence in the abstract into: The findings of this study could deepen our understanding of the physical process of permanent gully development from the perspective of hydrological and hydro-mechanics behavior of gully headcut.

Comment 2

Figure 1. Should be improved, China is floating in the white. In general, some items in the figures are difficult to read, for example legend items in Figure 1c, the location map bewteen 1b and 1c is impossible to see.

Reply 2

Done. We revised the figure 1. The new figure is shown as follows:

[Figure]

Fig. 2 Again some labels are very difficult to read, for example H in b.

Reply 3

Done.

Some labels in figure 2 are difficult to see. We improved the quality of the figure 2. The revised figure 2 is shown as follows:

[Figure]

Comment 4

L150 monitoring instead "observation work"

Reply 4

    OK. We already revised it.

Comment 5

Section 3.1 How many flights did you carry out? what was the monitoring period? dates, UAV type, resolution? I miss so many details here.

Reply 5

    Ok.

    We should give a clear description of the detailed information of the flights and monitoring period, dates, UAV type and the resolution of the digitized elevation model.

    In the revised manuscript, we clearly presented the missing information in the second paragraph of section 3.1

    To observe the gravitational mass-wasting process during the rainy and melting seasons, the study area was scanned by numerous control plates (the dots in Figs. 1a and 1b, and dashed circles in Figs. 2c and 2d), installed in and around the gully area and used unmanned aerial vehicle (Phantom 4 RTK-DJI). These control points were used to analyze the accuracy of the UAV-derived map and digital elevation model, aiming to obtain highly accurate topography. During the melting season in 2023 and the rainy season in 2022, three flights on 28 June 2022, 17 October 2022, and 20 June 2023, were carried out to obtain the high-resolution maps and digital elevation models (DEMs) with high resolutions of 0.058, 0.108 and 0.042m, respectively. The DEMs were spatially registered in ArcGIS 10.2 by a standard layer of orthoimage, ground control points, and the spline function. Then, the erosion depth of the headcut area could be obtained by the differences between two DEMs. Therefore, the linear and the areal erosion intensity can be calculated using the erosion depth and the grid size. Then, the differences between two digital elevation models generated the positive and negative terrain, which quantitatively showed the erosion intensity of the gravitational mass-wasting. Additionally, the eroded soil volume in the unit over-steepen slope surface area, termed areal erosion intensity, was applied in this work to address the erosion intensity of gravitational mass-wasting.

Comment 6

L213-219 This is methodology and should be placed there, for example in section 3.1

Reply 6

    Yes. We adopted your suggestion here. We revised the second paragraph in section 3.1: "To observe the gravitational mass-wasting process during the rainy and melting seasons, the study area was scanned by numerous control plates (the dots in Figs. 1a and 1b, and dashed circles in Figs. 2c and 2d), installed in and around the gully area and used unmanned aerial vehicle (Phantom 4 RTK-DJI). These control points were used to analyze the accuracy of the UAV-derived map and digital elevation model, aiming to obtain highly accurate topography. During the melting season in 2023 and the rainy season in 2022, three flights on 28 June 2022, 17 October 2022, and 20 June 2023, were carried out to obtain the high-resolution maps and digital elevation models (DEMs) with high resolutions of 0.058, 0.108 and 0.042m, respectively. The DEMs were spatially registered in ArcGIS 10.2 by a standard layer of orthoimage, ground control points, and the spline function. Then, the erosion depth of the headcut area could be obtained by the differences between two DEMs. Therefore, the linear and the areal erosion intensity can be calculated using the erosion depth and the grid size. Then, the differences between two digital elevation

models generated the positive and negative terrain, which quantitatively showed the erosion intensity of the gravitational mass-wasting. Additionally, the eroded soil volume in the unit over-steepen slope surface area, termed areal erosion intensity, was applied in this work to address the erosion intensity of gravitational mass-wasting."

Comment 7

RESULTS: there are many interpretations of the results that, in my opinion, are discussions more than objective results, therefore, I suggest to move these sentences to the corresponding section and use the results just for the objective introduction of observed data or processes.

Reply 7

Done.

We checked the results section and some paragraphs should be in the discussion part. Except some brief interpretations before a result section (we already moved to corresponding section, either in the method section or the discussion section), four paragraphs should be moved, they are "the second paragraph of 4.1 section, the third paragraph of 4.4 section, and the first and the fourth paragraph of 4.3.2 section. The 4.2 and 4.3 section clearly gave the objective introduction of the observed data and analysis.

The second paragraph of 4.1 section act a connecting role to the following result section. It gave a brief analysis about and clear explanation on the 4.2, 4.3 and 4.4 section. Of course, the first sentence with citations has been deleted because it is useless. Therefore, we suggest that keep the revised second paragraph of 4.1 section.

The third paragraph of 4.4 section cited a reference, aiming to compare the rainfall threshold. We revised this paragraph, and merely keep the objective results. For the rainfall threshold comparison, we moved them into the discussion section.

The first paragraph of 4.3.2 section repeated with some sentences in the discussion part and the fourth paragraph of 4.3.2 section. Therefore, he fourth paragraph of 4.3.2 section should be kept and the first paragraph of 4.3.2 section should be merged into the discussion section.

Comment 8

L241- These instead this

Reply 8

Done.

We already revised it.

Comment 9

L281 - Soil instead soi

Reply 9

Done.

We already revised it.

Comment 10

L271- I think you used this acronym before

Reply 10

Yes.

We used this acronym before, so we revised it into full text.

L315-139 - Again methodological issues in the results section.

Reply 11

    Done.

    We already moved it into the last paragraph of the methodology part.

Comment 12

L321- 0,15ºC per day? specify

Reply 12

    Yes. You're right here. Thanks for your reminding here.

    We already added "per day" after the number.

Comment 13

Fig. 8- I cannot see some details in (a), also happens in Fig. 9 (b)

Reply 13

    Done.

    We added a new figure to give a clear presentation of the figs. 8a and 9b.

[Figure]

The caption of the new figure is: **Fig. 9.** Hydrologic behavior for gully headcut during light rain events. (**a**) relatively lower rate of increasing of VWC for gully No. I. (**b**) relatively higher soil water storage for gully No. II. The three crossing lines of box show the 75th quantile ($Q_3$), median ($Q_2$), and 25th quantile ($Q_1$) from top to bottom. The length of the box is referred to as the interquartile range (IQR= $Q_3$- $Q_1$). The crossed square inside the box is the average value. The upper limit and lower limit of whiskers are $Q_3+1.5$IQR and $Q_3-1.5$IQR, respectively. The solid squares are the outliers.

Comment 14

L356-Avoid citations in the results.

Reply 14

    Done.

    We already deleted the citation in the results.

Comment 15

Fig. 10- the title of X-axis is not visible

Reply 15

Done.

I'm worry here to miss the title of X-axis in figure 10.

We already revised it.

[Figure]

Additionally, we added a new figure (corresponding to the comment 13)

The caption of the new figure is: **Fig. 9.** Hydrologic behavior for gully headcut during light rain events. (**a**) relatively lower rate of increasing of VWC for gully No. I. (**b**) relatively higher soil water storage for gully No. II. The three crossing lines of box show the 75th quantile ($Q_3$), median ($Q_2$), and 25th quantile ($Q_1$) from top to bottom. The length of the box is referred to as the interquartile range (IQR= $Q_3$- $Q_1$). The crossed square inside the box is the average value. The upper limit and lower limit of whiskers are $Q_3+1.5$IQR and $Q_3-1.5$IQR, respectively. The solid squares are the outliers.

---

## Author Comment (AC2)

**Brief comment**: This study observed erosion processes and some soil-hydrological properties at the headcut areas of two permanent gullies in Northeast China's Mollisols region during rainy and snow-melting seasons. Key parameters like soil moisture, temperature, and precipitation were investigated to understand water storage capacity, leakage processes, and suction stress levels. Although only two headcuts were monitored, I think the results could be of interest for the scientific community, but I would only recommend the publication of the work if a series of changes and improvements are carried out. In my opinion, the authors wrongly include methodological content and interpretations in the results section. These contents should be rightly placed in the corresponding section. Additionally, I miss some methodological details that I consider relevant (UAV data, processing, etc. see my comments below). Finally, figures must be notably improved, particularly the font size, please see my comments below. I detail these suggestions in the following lines:

**Replies:**

Thanks for your comments to improve the quality of this manuscript.

Firstly, the physical process of gravitational soil erosion in permanent gully has been long neglected in the soil-water conservation research field, while it plays an important role in permanent gully expansion and development. In this work, we clearly addressed their differences and similarities with the landslides and used the theory of unsaturated soil mechanics to study their physical process.

As you suggested, we should clearly state some methodological content and interpretations in the method part, not in the results part. Meanwhile, some short paragraph in the results part should be moved into discussion part. We also found that the texts in some figures are not clear enough to read and some errors exists. Therefore, we made a thorough revision of this manuscript and the figures.

We made a throughout revision for the previous manuscript. **Please see the manuscript with marked changes and accepted changes.**

We made a point-to-point response to your comments, as follows:

**Comment 1**

Abstract: I have doubts about the last sentence as the USLE was not designed to predict gully erosion. I would recommend to delete this sentence.

**Reply 1**

Done. We revised the last sentence of in the abstract. As you know, USLE is an empirically-based function. If a physically-based function is developed to predict the gully erosion, it will be better. We revised the last sentence in the abstract into: The findings of this study could deepen our understanding of the physical process of permanent gully development from the perspective of hydrological and hydro-mechanics behavior of gully head-cut.

**Comment 2**

Figure 1. Should be improved, China is floating in the white. In general, some items in the figures are difficult to read, for example legend items in Figure 1c, the location map between 1b and 1c is impossible to see.

**Reply 2**

Done.

The other two reviewers also mentioned the quality of figure 1.

We revised the figure 1. The new figure is shown as follows:

[Figure]

**Fig. 1.** Location of the two permanent gullies in the Mollisols region of northeast China. **(a)** The red star marks observation site in the study area (from ESRI). **(b)** Monitoring sites and ground controlling points at permanent gully No. I. **(c)** Monitoring sites and ground controlling points at permanent gully No. II. (background of **a** is from ESRI; areal maps of **b** and **c** are from UAV by Shoupeng Wang; the area between the blue lines mark gully bed, and the area between pink and blue lines mark the steep slope).

**Comment 3**

Fig. 2 Again some labels are very difficult to read, for example H in b.

**Reply 3**

Done.

Some labels in figure 2 are difficult to see. We improved the quality of the figure 2. The revised figure 2 is shown as follows:

[Figure]

**Fig. 2.** A close view of the over-steepen slope and headcut of the two permanent gullies, with **(a)** cross section and upstream view of the permanent gully No. I, **(b)** cross section and downstream view of the permanent gully No.

II, (c) ground controlling points (blue dot circles) and the soil moisture-temperature monitoring site (yellow star) at permanent gully No. I, and (d) ground controlling points and the soil moisture-temperature monitoring sites at permanent gully No. II. The location of headcut of the two gullies is shown in fig. 1. The area between blue lines marks the gully bed. The area between the pink and blue lines marks the slope.

**Comment 4**
L150 monitoring instead "observation work"
**Reply 4**
Done.

We already revised it.

**Comment 5**
Section 3.1 How many flights did you carry out? what was the monitoring period? dates, UAV type, resolution? I miss so many details here.
**Reply 5**
Ok.

The other two reviewers also mentioned what you said here. We also found that we missed the important flights during the study period. We should give a clear description of the detailed information of the flights and monitoring period, dates, UAV type and the resolution of the digitized elevation model.

In the revised manuscript, we clearly presented the missing information in the second paragraph of section 3.1: "To observe the gravitational mass-wasting process during the rainy and melting seasons, the study area was scanned by numerous control plates (the dots in Figs. 1a and 1b, and dashed circles in Figs. 2c and 2d), installed in and around the gully area and used unmanned aerial vehicle. These control points were used to analyze the accuracy of the UAV-derived map and digital elevation model, aiming to obtain highly accurate topography. During the melting season in 2023 and the rainy season in 2022, three flights on 28 June 2022, 17 October 2022, and 20 June 2023, were implemented with same flight routine and image overlap. We used Pix4D software to process image synthesis and the gully topography producing, which can reallocate the point cloud and filter the points of vegetation layer. As the points of vegetation layer (mainly the grass leaf) is changeable in plant height while the ground point is fixable, the vegetation layer could be filtered out and removed through the filtering tool. The DEMs products were spatially registered in ArcGIS 10.2 by a standard layer of orthoimage, ground control points, and the spline function (Table 1). Then, the erosion depth of the headcut area could be obtained by the differences between two DEMs. Therefore, the linear and the erosion intensity can be calculated using the erosion depth and the grid size. Then, the differences between two digital elevation models generated the positive and negative terrain, which quantitatively showed the erosion intensity of the gravitational mass-wasting. Additionally, the eroded soil volume in the unit over-steepen slope surface area, termed erosion per unit area, was applied in this work to address the erosion intensity of gravitational mass-wasting."

**Table 1.** Detailed information of three UAV flights and the digital elevation models

| UAV model | Flight date | Flight height (m) | DEM Accuracy (m) | Image overlap (%) |
|---|---|---|---|---|
| DJI Inspire 2 RTK | 2022.06.28 | 200 | 0.058 | 80 |
| DJI Phantom 4 RTK | 2022.10.17 | 500 | 0.108 | 80 |
| DJI Phantom 4 RTK | 2023.06.21 | 150 | 0.042 | 80 |

**Comment 6**
L213-219 This is methodology and should be placed there, for example in section 3.1

**Reply 6**

Yes. We adopted your suggestion here. We revised the second paragraph in section 3.1: "To observe the gravitational mass-wasting process during the rainy and melting seasons, the study area was scanned by numerous control plates (the dots in Figs. 1a and 1b, and dashed circles in Figs. 2c and 2d), installed in and around the gully area and used unmanned aerial vehicle. These control points were used to analyze the accuracy of the UAV-derived map and digital elevation model, aiming to obtain highly accurate topography. During the melting season in 2023 and the rainy season in 2022, three flights on 28 June 2022, 17 October 2022, and 20 June 2023, were implemented with same flight routine and image overlap. We used Pix4D software to process image synthesis and the gully topography producing, which can reallocate the point cloud and filter the points of vegetation layer. As the points of vegetation layer (mainly the grass leaf) is changeable in plant height while the ground point is fixable, the vegetation layer could be filtered out and removed through the filtering tool. The DEMs products were spatially registered in ArcGIS 10.2 by a standard layer of orthoimage, ground control points, and the spline function (Table 1). Then, the erosion depth of the headcut area could be obtained by the differences between two DEMs. Therefore, the linear and the erosion intensity can be calculated using the erosion depth and the grid size. Then, the differences between two digital elevation models generated the positive and negative terrain, which quantitatively showed the erosion intensity of the gravitational mass-wasting. Additionally, the eroded soil volume in the unit over-steepen slope surface area, termed erosion per unit area, was applied in this work to address the erosion intensity of gravitational mass-wasting."

**Table 1.** Detailed information of three UAV flights and the digital elevation models

| UAV model | Flight date | Flight height (m) | DEM Accuracy (m) | Image overlap (%) |
|---|---|---|---|---|
| DJI Inspire 2 RTK | 2022.06.28 | 200 | 0.058 | 80 |
| DJI Phantom 4 RTK | 2022.10.17 | 500 | 0.108 | 80 |
| DJI Phantom 4 RTK | 2023.06.21 | 150 | 0.042 | 80 |

**Comment 7**

RESULTS: there are many interpretations of the results that, in my opinion, are discussions more than objective results, therefore, I suggest to move these sentences to the corresponding section and use the results just for the objective introduction of observed data or processes.

**Reply 7**

Done.

We checked the results section and some paragraphs should be in the discussion part. Except some brief interpretations before a result section (we already moved to corresponding section, either in the method section or the discussion section), four paragraphs should be moved, they are "the second paragraph of 4.1 section, the third paragraph of 4.4 section, and the first and the fourth paragraph of 4.3.2 section. The 4.2 and 4.3 section clearly gave the objective introduction of the observed data and analysis.

The second paragraph of 4.1 section act a connecting role to the following result section. It gave a brief analysis about and clear explanation on the 4.2, 4.3 and 4.4 section. Of course, the first sentence with citations has been deleted because it is useless. Therefore, we suggest that keep the revised second paragraph of 4.1 section.

The third paragraph of 4.4 section cited a reference, aiming to compare the rainfall threshold. We revised this paragraph, and merely keep the objective results. For the rainfall threshold comparison, we moved them into the discussion section.

The first paragraph of 4.3.2 section repeated with some sentences in the discussion part and

the fourth paragraph of 4.3.2 section. Therefore, he fourth paragraph of 4.3.2 section should be kept and the first paragraph of 4.3.2 section should be merged into the discussion section.

**Comment 8**

L241- These instead this

**Reply 8**

Done. Thanks a lot.

We already revised it.

**Comment 9**

L281 - Soil instead soi

**Reply 9**

Done. Sorry to make such a mistake here. Thanks for your kind reminding.

We already revised it.

**Comment 10**

L271- I think you used this acronym before

**Reply 10**

Yes.

We used this acronym before, so we revised it into full text.

**Comment 11**

L315-139 - Again methodological issues in the results section.

**Reply 11**

Yes. After we read the previous manuscript again, we found some methodological issues should be in the results section. Meanwhile, we revised the methodology, results and discussions part.

We already moved it into the last paragraph of the methodology part.

**Comment 12**

L321- 0,15ºC per day? specify

**Reply 12**

Yes. You're right here. Thanks for your reminding here.

We already added "per day" after the number.

**Comment 13**

Fig. 8- I cannot see some details in (a), also happens in Fig. 9 (b)

**Reply 13**

Done.

We added a new figure to give a clear presentation of the figs. 8a and 9b.

[Figure]

The caption of the new figure is: **Fig. 9.** Hydrologic behavior for gully headcut during light rain events. (**a**) relatively lower rate of increasing of VWC for gully No. I. (**b**) relatively higher soil water storage for gully No. II. The three crossing lines of box show the 75th quantile ($Q_3$), median ($Q_2$), and 25th quantile ($Q_1$) from top to bottom. The length of the box is referred to as the interquartile range (IQR= $Q_3$- $Q_1$). The crossed square inside the box is the average value. The upper limit and lower limit of whiskers are $Q_3+1.5$IQR and $Q_3-1.5$IQR, respectively. The solid squares are the outliers.

**Comment 14**

L356-Avoid citations in the results.

**Reply 14**

Done. Results part should gave objective description of the founding.

We already deleted the citation in the results.

**Comment 15**

Fig. 10- the title of X-axis is not visible

**Reply 15**

Done.

I'm worry here to miss the title of X-axis in figure 10.

We already revised it. Besides, the other two reviewers mentioned me that the $R^2$ is not enough to support the significance of the fitted line. So, I inserted the P value in figs. 10a and 10b.

[Figure]

**Fig. 11.** Relationship between hydrology and the hydro-mechanical state with the erosion intensity. **(a)** Suction stress during the rainy season. **(b)** Suction stress during the snow-melting season. **(c)** erosion per unit area on over-steepen slope decreases with suction stress. **(d)** erosion per unit area on channel bed decreases with water storage amount.

**Comment 16**

Additionally, we added a new figure (corresponding to the comment 13)

[Figure]

The caption of the new figure is: **Fig. 9.** Hydrologic behavior for gully headcut during light rain events. **(a)** relatively lower rate of increasing of VWC for gully No. I. **(b)** relatively higher soil water storage for gully No. II. The three crossing lines of box show the 75th quantile ($Q_3$), median ($Q_2$), and 25th quantile ($Q_1$) from top to bottom. The length of the box is referred to as the interquartile range (IQR= $Q_3$- $Q_1$). The crossed square inside the box is the average value. The upper limit and lower limit of whiskers are $Q_3$+1.5IQR and $Q_3$−1.5IQR, respectively. The solid squares are the outliers.

---

## Author Comment (AC3)

**Comment:** The hydro-mechanical properties of soils were widely recognized as a major factor influencing some key sub-processes of gully erosion, but due to the difficulty of monitoring the infiltration during the gully development, especially in the field. Therefore, this study has important contribution to reveal the influences of different hydro-mechanical properties on gully erosion through a well design monitoring scheme on soil-related properties. But there are a few key questions need be explained clearer and precisely, which may strongly affect the results and conclusion of this study. I suggest major revision right now.

**Replies:** Thanks for your recognition for our works.

We made a throughout revision for the previous manuscript. **Please see the manuscript with marked changes and accepted changes.**

From the perspective of unsaturated soil mechanics, any soil failure results from the imbalance of the force or stress. In the research field of soil erosion, the gravitational mass movement shares the same mechanics of soil slips or landsliding while the scale of gravitational mass movement is smaller than slips and sliding, and it is difficult to monitor the interior stress status. Therefore, soil water status and the soil stress of the soil mass may give insightful knowledge about the permanent gully expansion, which is the focus of this work. In this study, we carried out a well-design monitoring scheme to obtain the soil water status and suction stress. Our results do contribute a lot to the knowledge about the physical process of permanent gully development. In the next, we will continue to extend our methods and give a clearer introduction to the gravitational soil loss in melting and rainy season.

**Comment:** The authors should give more details about the gully monitoring by the UAV, and the processes to calculate the variation of the gullies, and the accuracy of the monitoring methods. How you dealt with the effects of vegetation on morphological changes of gullies? I saw a lot of plants on your selected gully beds.

**Replies:** Done.

Thanks for your reminding here.

The plants are grass, not trees. If the plants are trees, it is very difficult to hanlding them by our Resisted Pix4D software.

The other two reviewers also gave me such comments about the UAV information. We should give a clear description of the UAV information and how we use software (Resisted Pix4D software) to diminish the effect of vegetation on morphological changes of gullies. In the revised manuscript, we added a new table (table 1 in the revised manuscript) to give the detailed information of the UAV and three flights. As our UAV have Real-time kinematic (RTK for abbreviation), the digital elevation models are high-resolution. Besides, we used Pix4D software to deal with the vegetation problem by generating point cloud and filtering tool.

The orthomosaic images and corresponding digital elevation models are shown as follows:

| Date: 2022.06.28 | Date: 2022.10.17 | Date: 2023.06.21 |
|---|---|---|

[Figure]

The new table is:

**Table 1.** Detailed information of three UAV flights and the digital elevation models

| UAV model | Flight date | Flight height (m) | DEM Accuracy (m) | Image overlap (%) |
|---|---|---|---|---|
| DJI Inspire 2 RTK | 2022.06.28 | 200 | 0.058 | 80 |
| DJI Phantom 4 RTK | 2022.10.17 | 500 | 0.108 | 80 |
| DJI Phantom 4 RTK | 2023.06.21 | 150 | 0.042 | 80 |

Vegetation processing involved the following steps:

Step 1. We used Pix4D software to process image synthesis and the gully topography producing, which can reallocate the point cloud and filter the points of vegetation layer. As the points of vegetation layer (mainly the grass leaf) is changeable in plant height while the ground point is fixable, the vegetation layer could be filtered out and removed through the filtering tool.

Step 2. Following manual screening to ensure the removal of any residual vegetation layer point clouds, the elevation data was regenerated, yielding a processed Digital Elevation Model (DEM) for the watershed.

Step 3. The erosion mainly occurs in the slope area and the gully bed area. For sites beyond the gully area, the topography change does not consider in our works as these sites are flatten and not in the gully area. Therefore, gully edges were delineated through visual interpretation of RGB optical images, with efforts made to exclude vegetation on the banks to the greatest extent possible.

The DEM was resampled to 0.10 m using ArcGIS 10.8 software. Ground control points were employed to perform local precise registration of the drone aerial imagery within ArcGIS 10.8, thereby minimizing errors in gully delineation. These ground control points were also utilized to enhance the accuracy of three DEMs.

**Comment**: The authors should clarify the concept of "gully beds" and "slopes" in this manuscript, which I suggest you to marked the location of "gully beds" and "slopes" on figure 1b and c. And to me, the location of "slope" is very difficult to be determined.

**Replies**: Done.

We should give a clearer symbol to distinguish the gully bed and slope. In the revised manuscript, we improved the quality of figure 1 and figure 2.

[Figure]

**Fig. 1.** Location of the two permanent gullies in the Mollisols region of northeast China. **(a)** The red star marks observation site in the study area (from ESRI). **(b)** Monitoring sites and ground controlling points at permanent gully No. I. **(c)** Monitoring sites and ground controlling points at permanent gully No. II. (background of **a** is from ESRI; areal maps of **b** and **c** are from UAV by Shoupeng Wang; the area between the blue lines mark gully bed, and the area between pink and blue lines mark the steep slope).

[Figure]

**Fig. 2.** A close view of the over-steepen slope and headcut of the two permanent gullies, with **(a)** cross section and upstream view of the permanent gully No. I, **(b)** cross section and downstream view of the permanent gully No. II, **(c)** ground controlling points (blue dot circles) and the soil moisture-temperature monitoring site (yellow star) at permanent gully No. I, and **(d)** ground controlling points and the soil moisture-temperature monitoring

sites at permanent gully No. II. The location of headcut of the two gullies is shown in fig. 1. The area between blue lines marks the gully bed. The area between the pink and blue lines marks the slope.

**Comment:** The discussion part is quite weak right now. The author should compared your results with previous studies, especially some studies related to gully piping erosion. Soil properties i.e bulk density, grain sizes and porosity have great influence on soil hydro-mechanical properties, and more easy to obtain the data. So I suggest the authors to analyze the relationship between these properties.

**Replies:** Done.

Maybe you are good at the pipping erosion, which is one of the most important factors causing permanent gully development. Discussion part in the previous manuscript is a little weak and the quality must be improved. In the revised manuscript, we added some sentences to mention what your good suggestions here.

Previously, we found that the research about permanent gully expansion or development mainly focus on the runoff process or topographical threshold, and neglects the role of soil stress. The gravitational mass movement, soil slips, avalanches (merely soil water-related) are closely related to the soil stress status, which is the main ideology of this work. Besides, this work used complicated and expensive method to obtain the soil hydro-mechanical properties. We nearly used 14 days to measure the wetting and drying process (The reason that such long duration lies in the fine particles of Mollisols). Then, we derived the soil-water-characteristic-curve using Hydrus 1-D (may cost me another 14 days). As you said, bulk density, grain sizes and porosity have great influence on soil hydro-mechanical properties, and easier to obtain them. However, these data must be tested by the accuracy apparatus, and could be used for soil hydro-mechanical property analysis. For example, the soil porosity, can be divided into matrix pore and macro pore. It has various class by the pore size and which pore determines the hydro-mechanical property is the greatest challenge in Unsaturated soil mechanics. The reason why we choose the two gullies lie in that the soils are undisturbed and disturbed. Such soil status would result in the differences in the hydro-mechanical properties. Therefore, we wrote some sentences in the revised manuscript to highlight this aspect.

**Comment:** The erosion of "slope" was correlated with suction stress is OK to me (Fig 10.c), but the "gully bed" normally considered to eroded surface runoff, and how to explain the influences by water storage need be more clear.

**Replies:** Done.

Erosion of gully bed positively relates to surface runoff. In a given rain event, the more surface runoff, the less the water storage. In fact, water storage and the runoff depth were approximately equal to the rainfall depth. Consequently, the erosion per unit area of the channel bed was inversely proportional to the water storage.

In the revised manuscript, figure 11c (as figure 10c in previous manuscript) sufficient prove that the low suction stress (or high soil moisture) corresponds to the high slope erosion or intensive gravitational mass movement. Figure 11d describes the stored

water with the erosion. The more the water is stored in rain events, the less the runoff a catchment will produce.

In the revised manuscript, we strengthened this part in the third paragraph of Discussion.

**Comment**: Line 27: "gully bead"?

**Replies**: Sorry to make a mistake here.

It should be gully bed, not gully bead. We already revised it.

**Comment**: Line 49-50: "one of the most important factors in the development of permanent gullies, could be determined by the topographical threshold and volumetric retreat rate of gully headcut", the development of gullies determined by three main processes: headcut retreat, deepen and widen.

**Replies**: Thanks for your exact definition of permanent gully development.

We know that gully development derives from retreat, deepen and widen. We should write a more exact description about the gully development here.

In my opinion, gully deepen mainly relates to the concentrated runoff water. Widen refers to the gravitational mass movement from gully slope. Headcut retreat derives from the deepen process. This sentence seems to be a little redundance. So I revised it into "Permanent gullies initiate in locations where concentrated flow can erode and delivery bed sediments (Kirkby and Bracken, 2009), and expand at the over-steepen slopes when gravitational mass-wasting process occur following instant or constant water infiltration (Poesen et al., 2010; Tebebu et al., 2010). Development of permanent gullies, could be determined by the topographical threshold and volumetric retreat rate of gully headcut (Svoray et al., 2012; Guan et al., 2021; Zare et al., 2022), gully length-area-volume relationship (Li et al., 2015 and 2017), as well as their function with upstream drainage area and rainy days in different environments (Hayas et al., 2019)."

**Comment**: Line 56-57 "most studies on permanent gullies have primarily concentrated on the gully headcut retreat and topographic threshold conditions", this sentence did not summarized the previous studies well, I suggest that the authors emphasize the GHR processes through surface runoff processes as previous studies (contrast with runoff infiltration), and delete the topographic threshold part, which is not directly related to this study.

**Replies**: Done.

Thanks for your suggestion here. The same problem with the Line 49-50.

I have to say that tremendous GHR studies in the world and we should improve our knowledge about the GHR through surface runoff process.

Follow your suggestion here, we deleted line 56-57 as this sentence seems to be not useful and is not helpful for the contents of this work.

**Comment**: Line 114: check the total area of mollisols regions, which normally over 1,000 000 km$^2$.

**Replies**: Sorry to make a mistake here.

It should be 1,030 000 km$^2$. We already revised it.

**Comment:** Line 115: whether maize belong to grain or not?

**Replies:** We timely checked it.

We revised the maize into corn.

Of course, if you have more detailed description here, we would like to accept your suggestions.

**Comment:** Line 125-127: give some basic topographic parameters of these two gullies i.e length, widths, depths, area, volumes.

**Replies:** Done.

Sorry to miss your mentioned some gully information here.

In previous manuscript, we merely give the geometric of the headcut area of the two gullies. In revised manuscript, we

In the revised manuscript, we used one paragraph to describe the basic topographic parameters of the two gullies: "The observed two permanent gullies in this work are 1.4 km apart and are located on south-facing and north-facing rolling-slope respectively (Figs. 1b and 1c). The catchment area above the heacut of gully No. I is 0.22 km$^2$. The relative relief and the channel gradient are 25.85m and 3.3%. The catchment above the headcut of gully No. II is 0.35 km$^2$, and the relative relief and channel gradient are 26.1 m and 3.2%. The width of gully No. I gradually broadens while that of gully No. II becomes narrow, and the depth of gully No. I is deeper (Figs. 2a and 2b). In detail, the mean depth of the gully No. I is 3.5m while that of gully No. II is 1.23 m. The mean length and width of No. I gully are 25.3m and 8.72m, while those of gully No. II are 28.2 and 5.61 m. The gully area for No. I is 199.3 m$^2$ and the volume is 863.6 m$^3$. For gully No. II, the gully area and volume are 143.3 m$^2$ and 123.6 m$^3$."

**Comment:** Line 134-135: "The study area has a cold temperate continental monsoon climate with variable annual precipitations ranging from 480 mm to 512 mm, and 600 mm on average", confusion data.

**Replies:** Sorry to make a mistake here.

We checked the data with my teammates in the author list.

It should be "The study area has a continental monsoon climate with variable annual precipitation ranging from 347 mm to 775 mm, with 546 mm on average for the years of 1971 through 2018 (Tang et al. 2023)".

We already revised it in the revised manuscript.

**Comment:** The second paragraph of section 3.1: more details about the gully monitoring are required: control plates were only applied to check the accuracy of the UAV DSMs, or also as the ground control points to improve the accuracy of DSMs? Which software you used to produce the DSMs? What about the flight height and the overlaps of photos? How to reduce the influences of the vegetation in gullies.

**Replies:** Done.

Thanks for your reminding and another reviewer's comments about the UAV information. We added corresponding information and the strengthened this part in the revised manuscript.

The ground control points (GCPs) offer reference points enabling photogrammetry software to more accurately calibrate camera positions and orientations, as well as better

align and georeference aerial images. This step is crucial for ensuring the accuracy and correct spatial referencing of the DEM, particularly in applications requiring high-precision terrain modeling. We should give a clear description of the UAV information and how we use software (Resisted Pix4D software) to diminish the effect of vegetation on morphological changes of gullies. In the revised manuscript, we added a new table (table 1 in the revised manuscript) to give the detailed information of the UAV and three flights. As our UAV have Real-time kinematic (RTK for abbreviation), the digital elevation models are high-resolution. Besides, we used Pix4D software to deal with the vegetation problem by generating point cloud and filtering tool.

The orthomosaic images and corresponding digital elevation models are shown as follows:

| Date: 2022.06.28 | Date: 2022.10.17 | Date: 2023.06.21 |
| --- | --- | --- |

[Figure]

The new table is:

**Table 1.** Detailed information of three UAV flights and the digital elevation models

| UAV model | Flight date | Flight height (m) | DEM Accuracy (m) | Image overlap (%) |
| --- | --- | --- | --- | --- |
| DJI Inspire 2 RTK | 2022.06.28 | 200 | 0.058 | 80 |
| DJI Phantom 4 RTK | 2022.10.17 | 500 | 0.108 | 80 |
| DJI Phantom 4 RTK | 2023.06.21 | 150 | 0.042 | 80 |

Vegetation processing involved the following steps:

Step 1. We used Pix4D software to process image synthesis and the gully topography producing, which can reallocate the point cloud and filter the points of vegetation layer. As the points of vegetation layer (mainly the grass leaf) is changeable in plant height while the ground point is fixable, the vegetation layer could be filtered out and removed through the filtering tool.

Step 2. Following manual screening to ensure the removal of any residual vegetation layer point clouds, the elevation data was regenerated, yielding a processed Digital Elevation Model (DEM) for the watershed.

Step 3. The erosion mainly occurs in the slope area and the gully bed area. For sites beyond the gully area, the topography change does not consider in our works as these sites are flatten and not in the gully area. Therefore, gully edges were delineated through visual interpretation of RGB optical images, with efforts made to exclude vegetation on the banks to the greatest extent possible.

The DEM was resampled to 0.10 m using ArcGIS 10.8 software. Ground control points were employed to perform local precise registration of the drone aerial imagery within ArcGIS 10.8, thereby minimizing errors in gully delineation. These ground control points were also utilized to enhance the accuracy of three DEMs.

**Comment:** Fig 2 need a figure to show the locations of the two gullies in the catchment.
**Replies:** Done.

 The gully head of the two gullies are merely 1.4 km. In the revised process, we updated the location of the two gullies, and texted them in figure 1.

[Figure]

**Comment:** Fig 4. "Areal erosion intensity" is very confusion, I guess your means the volumetric changes divided by the area of the locations. If so, just "Erosion per unit area" is better. Also in the text.
**Replies:** Done.

 We adopt your good suggestion here.

 We modified the areal erosion intensity to erosion per unit area in figure 4 and figure 11. Also, all of the "areal erosion intensity" in the text (including the figure caption) were modified to "erosion per unit area".

[Figure]

**Fig. 4.** Differences in the erosion per unit area for gully bed and over-steepen slope

[Figure]

**Fig. 11.** Relationship between hydrology and the hydro-mechanical state with the erosion intensity. **(a)** Suction stress during the rainy season. **(b)** Suction stress during the snow-melting season. **(c)** erosion per unit area on over-steepen slope decreases with suction stress. **(d)** erosion per unit area on channel bed decreases with water storage amount.

---

## Author Comment (AC4)

**Comment:** This study explained the erosion intensity of gravitational mass-wasting at gully head by soil water storage and drainage, and the suction stress etc. Two permanent gullies were selected and studied in Northeast China's Mollisols region. It is interesting that understanding permanent gully expansion from both classical mechanics and the state of stress perspectives. However, I don't think this paper could be published before major reversion.

**Replies: Jianjun, Thanks for your valuable comments.**

Thanks for your recognitions for our works, this manuscript needs a throughout revision and we followed most of the comments from you and other two reviewers.

Also, we made a throughout revision for the previous manuscript. **Please see the manuscript with marked changes and accepted changes.**

This work firstly examines the soil loss at headcut area of permanent gully in the Mollisols region of Northeast China. Currently, no one addresses the hydro-mechanical properties of mollisols and find the relationship between erosion per unit area and the soil suction stress and water storage. The occurrence of gravitational mass movement share similar mechanics as soil slips and avalanches while their scale are too small to monitor. Therefore, the only and feasible way to know about their occurrence and erosion intensity should combine the hydrological monitoring and the unsaturated soil mechanics. This work could give the physical process and the mechanics of gravitational mass movement. Therefore, we believe that our work would contribute greatly to the study field of soil erosion, attract more focus for scholars in the soil-water-conservation, and deserves to be published.

**What I concerned is that**

**Comment:** method part: the erosion volume derived from the UAV images based on the DoD (difference of DEM). DoD, as a typical method in 2D, may lead to the large uncertainty on the slopes, especially on the over-steepen slopes. For example, Fig. 3 in Kang, H., Wang, W., Guo, M., Li, J., & Shi, Q. (2021). How does land use/cover influence gully head retreat rates? An in-situ simulation experiment of rainfall and upstream inflow in the gullied loess region, China. *Land Degradation & Development, 32*(9), 2789-2804. In this case, DoD was suitable for Fig. 3a condition, but not for Fig. 3b. Both conditions are typically at the gully head. The solution includes but not limited: M3C2 (Gao, C., Li, P., Hu, J., Yan, L., Latifi, H., Yao, W., Hao, M., Gao, J., Dang, T., & Zhang, S. (2021). Development of gully erosion processes: A 3D investigation based on field scouring experiments and laser scanning. Remote Sensing of Environment, 265, 112683). Furthermore, we noticed there were many vegetation in the gully, which may also result in large uncertainties in volume calculation because vegetation obscures the ground information that reflect erosion truly. If a LiDAR UAV with your used in this study, its ok. Please clarify this part in the methods.

**Replies:** Good and professional comments here.

Firstly, DoD, e.g., the difference of DEM, is a 3D (it has *x*, *y*, and *z* orientations, not 2D) method to find the topography change. Uncertainty (or errors of DEM) on the slopes is a problem in finding the topography change if there were no fixed objects (aiming for image and DEM registration) in the images. Therefore, we used lots of fixed control points to minimize the uncertainties. Meanwhile, we used same flight routine (flight direction is vertical to the channel descending) and overlap ratio to cover the headcut area. Therefore, three aspects, fixed control points, same flight routine, flight direct to channel descending, were used to ensure the topography data accuracy.

Secondly, what you said that the DoD method performs well under certain conditions (such as Figure 3a), yet may be limited in others (such as Figure 3b), particularly in complex topographies like gully heads. This fact is true in your study area (may be gravitational mass movement in the headcut of gully in Loess Plateau), but not so in this work. In fact, the reason why two gullies were chosen for study in this work lies in that their cross section is not reverse V type. If the cross section exhibits reverse V type, we will establish a bending stability model to examine their occurrence, not the hydro-mechanical method. Importantly, you can see the channel cross section of figure 3 that the cross section exhibits V to U type, not the reverse V type. As you know, the reverse V type would generate some shadow area (as far as I know from field investigations in Loess Plateau) and the topography at the steep slope cannot be obtained, but can be handled by Lidar.

[Figure]

Figure 4 clearly shows the V or U cross section profile, not the reverse V type.

Thirdly, we used Pix4D software to process image synthesis and the gully topography producing, which can reallocate the point cloud and filter the points of vegetation layer. As the points of vegetation layer (mainly the grass leaf) is changeable in plant height while the ground point is fixable, the vegetation layer could be filtered out and removed through the filtering tool. Following manual screening to ensure the removal of any residual vegetation

layer point clouds, the elevation data was regenerated, yielding a processed Digital Elevation Model (DEM) for the watershed.

The orthomosaic images and corresponding digital elevation models are shown as follows:

| Date: 2022.06.28 | Date: 2022.10.17 | Date: 2023.06.21 |
|---|---|---|
|
[Figure]
 | | |

The new table is:

**Table 1.** Detailed information of three UAV flights and the digital elevation models

| UAV model | Flight date | Flight height (m) | DEM Accuracy (m) | Image overlap (%) |
|---|---|---|---|---|
| DJI Inspire 2 RTK | 2022.06.28 | 200 | 0.058 | 80 |
| DJI Phantom 4 RTK | 2022.10.17 | 500 | 0.108 | 80 |
| DJI Phantom 4 RTK | 2023.06.21 | 150 | 0.042 | 80 |

Vegetation processing involved the following steps:

Step 1. We used Pix4D software to process image synthesis and the gully topography producing, which can reallocate the point cloud and filter the points of vegetation layer. As the points of vegetation layer (mainly the grass leaf) is changeable in plant height while the ground point is fixable, the vegetation layer could be filtered out and removed through the filtering tool.

Step 2. Following manual screening to ensure the removal of any residual vegetation layer point clouds, the elevation data was regenerated, yielding a processed Digital Elevation Model (DEM) for the watershed.

Step 3. The erosion mainly occurs in the slope area and the gully bed area. For sites beyond the gully area, the topography change does not consider in our works as these sites are flatten and not in the gully area. Therefore, gully edges were delineated through visual interpretation of RGB optical images, with efforts made to exclude vegetation on the banks to the greatest extent possible.

The DEM was resampled to 0.10 m using ArcGIS 10.8 software. Ground control points were employed to perform local precise registration of the drone aerial imagery within ArcGIS 10.8, thereby minimizing errors in gully delineation. These ground control points were also utilized to enhance the accuracy of three DEMs.

**Comment:** Paper was not well organized. Method part was not completed, missing UAV, remote sensing data processing and erosion volume calculation, some figures. Many paragraphs

in results part should in method or discussion part. Discussion part should be in serval sanctions corresponding to the results part.

**Replies:** Done

After we read the comments provided by you and the other two reviewers, we found that three aspects should be improved: first, the missing UAV information; second, some part should be moved to method or discussion part; third, strength the discussion part. We are deeply grateful for your professional feedback. We already make a throughout revision for the previous manuscript.

**Comment:** This paper mentioned rainy season and snow-melting season, and erosion intensity in different seasons was analyzed (Fig. 4). But this paper not analyzed how factors influencing erosion intensity in different season. In other words, whether soil properties, hydrology, soil water have variation between seasons? This issue is critical to the whole story.

**Replies:** Good suggestion and comment here.

We should strengthen what you said in the discussion part.

We added a discussion sentence to address what you said in the third paragraph of Discussion: "The headcut of gully No. II is greatly disturbed, which may result in higher permeability, quicker water pressure response, higher soil moisture status either in rainy season or snow-melting season. Meanwhile, the soil suction stress is lower and slope erosion is more intensity than those of gully No. I. Note that the distance between the two gullies is merely 1.4 km and the climatic conditions is similar. Therefore, it seems that the soil properties may be the dominant intrinsic factors governing erosion intensity of gully slope."

There are some specific suggestions below:

**Comment:** The place name in Fig. 1 Haerbin should be Harbin, right? And the xing'an range, is the same with L120 Khingan Mountains?

**Replies:** Done.

Thanks for your patience and recommendations here. It greatly improves the quality of our text, figures and tables.

You're right here. Haerbin should be Harbin

Xing'an range is same with Khingan Mountains

[Figure]

**Fig. 1.** Location of the two permanent gullies in the Mollisols region of northeast China. **(a)** The red star marks observation site in the study area (from ESRI). **(b)** Monitoring sites and ground controlling points at permanent gully No. I. **(c)** Monitoring sites and ground controlling points at permanent gully No. II. (background of **a** is from ESRI; areal maps of **b** and **c** are from UAV by Shoupeng Wang; the area between the blue lines mark gully bed, and the area between pink and blue lines mark the steep slope).

**Comment:** L132-133 both number was I?

**Replies:** Sorry to make a mistake here.

We revised it into "Moreover, the height and width of gully No. II are lower than those of gully No. I (Fig. 3), and the head-cut area of gully No. II experienced tillage activities, while the headcut area of gully No. I does not."

**Comment:** L181-182 Formatting

**Replies**: Sorry to make a mistake here.

We revised it "$b_\downarrow$ is the dissipation proxy reflecting the water drainage ability of soil mass at given confining pressure, and reflects the concavity of the pore water pressure dissipation curve."

**Comment:** L216-219 should be method

**Replies:** Yes.

We already moved them in the method part.

**Comment:** L223 bed area? Slope area?

**Replies:** Ok.

We should write a clear description about the erosion per unit area.

In the revised manuscript, we wrote a clear description about the erosion per unit area by paragraph of rainy season and snow-melting season.

The revised parts are shown as follows (three paragraphs):

The erosion per unit area in both bed and slope area in the snow-melting season for gully No. I was greater than that in gully No. II (Fig. 4), which could be driven by the low melting water storage and high melting water runoff at the headcut of gully No. I.

In the rainy season, the erosion per unit area for bed of No. II gully was notably greater than that in gully No. I, which may result from the rapid water storage and leakage, producing intensive runoff at the headcut of gully No. II. The erosion of the over-steepen slope was mainly from the gravitational mass-wasting process. For gully No. II, the erosion per unit area in the snow-melting season was significantly greater compared to that in the rainy season.

In the snow-melting season, the erosion per unit area for slope of No. II gully was greater than that in gully No. I. Though the erosion per unit area in the rainy season for gully No. I was higher than that for gully No. II, the difference was as negligible as that in snow-melting season. It is important to note that the slopes of the permanent gully were over-steepen, and the stability of the slope primarily depended on the soil suction stress, as a function of the hydro-mechanical properties and the soil moisture."

**Comment:** L234 discussion?

**Replies:** Done.

We already moved them into the discussion part.

**Comment:** L285 method?

**Replies:** No.

It should be in the result part, not in the method part.

Line 285 in previous manuscript shows the contents of parameters describing the soil and water characteristic curve (SWCC) and the hydraulic conductivity function (HCF).

**Comment:** L290-297 method?

**Replies:** Done.

It should be in the method part (3.3 Hydro-mechanical property).

**Comment:** In Figure 10, the sample size is too small, so the R square is meaningless, and the significance p-value should be shown.

**Replies:** Done.

Thanks for your comments and suggestions here. we added the R square and the significance p-value in the figure 10 (figure 11 in the revised manuscript).

[Figure]

**Fig. 11.** Relationship between hydrology and the hydro-mechanical state with the erosion intensity. **(a)** Suction stress during the rainy season. **(b)** Suction stress during the snow-melting season. **(c)** erosion per unit area on over-steepen slope decreases with suction stress. **(d)** erosion per unit area on channel bed decreases with water storage amount.